# Rethinking Generative Mode Coverage:
# A Pointwise Guaranteed Approach

**Peilin Zhong**[*]    **Yuchen Mo**[*]    **Chang Xiao**[*]    **Pengyu Chen**    **Changxi Zheng**

Columbia University
{peilin, chang, cxz}@cs.columbia.edu
{yuchen.mo, pengyu.chen}@columbia.edu

## Abstract

Many generative models have to combat *missing modes*. The conventional wisdom to this end is by reducing through training a statistical distance (such as $f$-divergence) between the generated distribution and provided data distribution. But this is more of a heuristic than a guarantee. The statistical distance measures a *global*, but not *local*, similarity between two distributions. Even if it is small, it does not imply a plausible mode coverage. Rethinking this problem from a game-theoretic perspective, we show that a complete mode coverage is firmly attainable. If a generative model can approximate a data distribution moderately well under a global statistical distance measure, then we will be able to find a mixture of generators that collectively covers *every* data point and thus *every* mode, with a lower-bounded generation probability. Constructing the generator mixture has a connection to the multiplicative weights update rule, upon which we propose our algorithm. We prove that our algorithm guarantees complete mode coverage. And our experiments on real and synthetic datasets confirm better mode coverage over recent approaches, ones that also use generator mixtures but rely on global statistical distances.

## 1   Introduction

A major pillar of machine learning, the *generative* approach aims at learning a data distribution from a provided training dataset. While strikingly successful, many generative models suffer from *missing modes*. Even after a painstaking training process, the generated samples represent only a limited subset of the modes in the target data distribution, yielding a much lower entropy distribution.

Behind the missing mode problem is the conventional wisdom of training a generative model. Formulated as an optimization problem, the training process reduces a statistical distance between the generated distribution and the target data distribution. The statistical distance, such as $f$-divergence or Wasserstein distance, is often a *global* measure. It evaluates an integral of the discrepancy between two distributions over the data space (or a summation over a discrete dataset). In practice, reducing the global statistical distance to a perfect zero is virtually a mission impossible. Yet a small statistical distance does not certify the generator complete mode coverage. The generator may neglect underrepresented modes—ones that are less frequent in data space—in exchange for better matching the distribution of well represented modes, thereby lowering the statistical distance. In short, a global statistical distance is not ideal for promoting mode coverage (see Figure 1 for a 1D motivating example and later Figure 2 for examples of a few classic generative models).

This inherent limitation is evident in various types of generative models (see Appendix A for the analysis of a few classic generative models). Particularly in generative adversarial networks (GANs), mode collapse has been known as a prominent issue. Despite a number of recent improvements toward alleviating it [1, 2, 3, 4, 5, 6], none of them offers a complete mode coverage. In fact, even the

---

[*]equal contribution

fundamental question remains unanswered: *what precisely does a complete mode coverage mean?* After all, the definition of "modes" in a dataset is rather vague, depending on what specific distance metric is used for clustering data items (as discussed and illustrated in [4]).

We introduce an explicit notion of complete mode coverage, by switching from the global statistical distance to local *pointwise* coverage: provided a target data distribution $P$ with a probability density $p(x)$ at each point $x$ of the data space $\mathcal{X}$, we claim that a generator $G$ has a *complete mode coverage* of $P$ if the generator's probability $g(x)$ for generating $x$ is pointwise lower bounded, that is,

$$g(x) \geq \psi \cdot p(x), \forall x \in \mathcal{X}, \tag{1}$$

for a reasonably large relaxation constant $\psi \in (0, 1)$. This notion of mode coverage ensures that *every point $x$* in the data space $\mathcal{X}$ will be generated by $G$ with a finite and lower-bounded probability $g(x)$. Thereby, in contrast to the generator trained by reducing a global statistical distance (recall Figure 1), no mode will have an arbitrarily small generation probability, and thus no mode will be missed. Meanwhile, our mode coverage notion (1) stays compatible with the conventional heuristic toward reducing a global statistical distance, as the satisfaction of (1) implies that the total variation distance between $P$ and $G$ is upper bounded by $1 - \psi$ (see a proof in Appendix C).

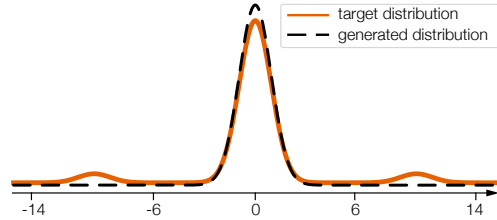

Figure 1: **Motivating example.** Consider a 1D target distribution $P$ with three modes, i.e., a mixture of three Gaussians, $P = 0.9 \cdot \mathcal{N}(0, 1) + 0.05 \cdot \mathcal{N}(10, 1) + 0.05 \cdot \mathcal{N}(-10, 1)$ (solid orange curve). If we learn this distribution using a single Gaussian $Q$ (black dashed curve). The statistical distance between the two is small: $D_{\text{TV}}(Q \parallel P) \leq 0.1$ and $D_{\text{KL}}(Q \parallel P) \leq 0.16$. The probability of drawing samples from the side modes (in $[-14, -6]$ and $[6, 14]$) of the target distribution $P$ is $\Pr_{x \sim P}[6 \leq |x| \leq 14] \approx 0.1$, but the probability of generating samples from $Q$ in the same intervals is $\Pr_{x \sim Q}[6 \leq |x| \leq 14] \approx 10^{-9}$. The side modes are missed!

At first sight, the pointwise condition (1) seems more stringent than reducing a global statistical distance, and pursuing it might require a new formulation of generative models. Perhaps somewhat surprisingly, a rethink from a game-theoretic perspective reveal that this notion of mode coverage is viable without formulating any new models. Indeed, a mixture of existing generative models (such as GANs) suffices. In this work, we provide an algorithm for constructing the generator mixture and a theoretical analysis showing the guarantee of our mode coverage notion (1).

## 1.1 A Game-Theoretic Analysis

Before delving into our algorithm, we offer an intuitive view of *why our mode coverage notion* (1) *is attainable* through a game-theoretic lens. Consider a two-player game between Alice and Bob: given a target data distribution $P$ and a family $\mathcal{G}$ of generators[2], Alice chooses a generator $G \in \mathcal{G}$, and Bob chooses a data point $x \in \mathcal{X}$. If the probability density $g(x)$ of Alice's $G$ generating Bob's choice of $x$ satisfies $g(x) \geq \frac{1}{4} p(x)$, the game produces a value $v(G, x) = 1$, otherwise it produces $v(G, x) = 0$. Here $1/4$ is used purposely as an example to concretize our intuition. Alice's goal is to maximize the game value, while Bob's goal is to minimize the game value.

Now, consider two situations. In the first situation, Bob first chooses a mixed strategy, that is, a distribution $Q$ over $\mathcal{X}$. Then, Alice chooses the best generator $G \in \mathcal{G}$ according to Bob's distribution $Q$. When the game starts, Bob samples a point $x$ using his choice of distribution $Q$. Together with Alice's choice $G$, the game produces a value. Since $x$ is now a random variable over $Q$, the expected game value is $\max_{G \in \mathcal{G}} \mathbb{E}_{x \sim Q}[v(G, x)]$. In the second situation, Alice first chooses a mixed strategy, that is, a distribution $R_\mathcal{G}$ of generators over $\mathcal{G}$. Then, given Alice's choice $R_\mathcal{G}$, Bob chooses the best data point $x \in \mathcal{X}$. When the game starts, Alice samples a generator $G$ from the chosen distribution $R_\mathcal{G}$. Together with Bob's choice of $x$, the game produces a value, and the expected value is $\min_{x \in \mathcal{X}} \mathbb{E}_{G \sim R_\mathcal{G}}[v(G, x)]$.

According to von Neumann's minimax theorem [7, 8], Bob's optimal expected value in the first situation must be the same as Alice's optimal value in the second situation:

$$\min_{Q} \max_{G \in \mathcal{G}} \mathbb{E}_{x \sim Q}[v(G, x)] = \max_{R_\mathcal{G}} \min_{x \in \mathcal{X}} \mathbb{E}_{G \sim R_\mathcal{G}}[v(G, x)]. \tag{2}$$

With this equality realized, our agenda in the rest of the analysis is as follows. First, we show a lower bound of the left-hand side of (2), and then we use the right-hand side to reach the lower-bound of $g(x)$ as in (1), for Alice's generator $G$. To this end, we need to depart off from the current game-theoretic analysis and discuss the properties of existing generative models for a moment.

Existing generative models such as GANs [9, 1, 10] aim to reproduce arbitrary data distributions. While it remains intractable to have the generated distribution match *exactly* the data distribution, the approximations are often plausible. One reason behind the plausible performance is that the data space encountered in practice is "natural" and restricted—all English sentences or all natural object images or all images on a manifold—but not a space of arbitrary data. Therefore, it is reasonable to expect the generators in $\mathcal{G}$ (e.g., all GANs) to meet the following requirement[3] (without conflicting the no-free-lunch theorem [11]): for any distribution $Q$ over a natural data space $\mathcal{X}$ encountered in practice, there exists a generator $G \in \mathcal{G}$ such that the total variation distance between $G$ and $Q$ is upper bounded by a constant $\gamma$, that is, $\frac{1}{2} \int_{\mathcal{X}} |q(x) - g(x)| \, dx \leq \gamma$, where $q(\cdot)$ and $g(\cdot)$ are the probability densities on $Q$ and the generated samples of $G$, respectively. Again as a concrete example, we use $\gamma = 0.1$. With this property in mind, we now go back to our game-theoretic analysis.

Back to the first situation described above. Once Bob's distribution $Q$ (over $\mathcal{X}$) and Alice's generator $G$ are identified, then given a target distribution $P$ over $\mathcal{X}$ and an $x$ drawn by Bob from $Q$, the probability of having Alice's $G$ cover $P$ (i.e., $g(x) \geq \frac{1}{4}p(x)$) at $x$ is lower bounded. In our current example, we have the following lower bound:

$$\Pr_{x \sim Q} [g(x) \geq 1/4 \cdot p(x)] \geq 0.4. \tag{3}$$

Here $0.4$ is related to the total variation distance bound (i.e., $\gamma = 0.1$) between $G$ and $Q$, and this lower bound value is derived in Appendix D. Next, notice that on the left-hand side of (2), the expected value, $\mathbb{E}_{x \sim Q}[v(G, x)]$, is equivalent to the probability in (3). Thus, we have

$$\min_Q \max_{G \in \mathcal{G}} \mathbb{E}_{x \sim Q}[v(G, x)] \geq 0.4. \tag{4}$$

Because of the equality in (2), this is also the lower bound of its right-hand side, from which we know that there exists a distribution $R_{\mathcal{G}}$ of generators such that for any $x \in \mathcal{X}$, we have

$$\mathbb{E}_{G \sim R_{\mathcal{G}}}[v(G, x)] = \Pr_{G \sim R_{\mathcal{G}}} [g(x) \geq 1/4 \cdot p(x)] \geq 0.4. \tag{5}$$

This expression shows that for any $x \in \mathcal{X}$, if we draw a generator $G$ from $R_{\mathcal{G}}$, then with a probability at least $0.4$, $G$'s generation probability density satisfies $g(x) \geq \frac{1}{4}p(x)$. Thus, we can think $R_{\mathcal{G}}$ as a "collective" generator $\boldsymbol{G}^*$, or a *mixture of generators*. When generating a sample $x$, we first choose a generator $G$ according to $R_{\mathcal{G}}$ and then sample an $x$ using $G$. The overall probability $g^*(x)$ of generating $x$ satisfies $g^*(x) > 0.1p(x)$—precisely the pointwise lower bound that we pose in (1).

**Takeaway from the analysis.** This analysis reveals that a complete mode coverage is firmly viable. Yet it offers no recipe on *how* to construct the mixture of generators and their distribution $R_{\mathcal{G}}$ using existing generative models. Interestingly, as pointed out by Arora et al. [12], a constructive version of von Neumann's minimax theorem is related to the general idea of multiplicative weights update. Therefore, our key contributions in this work are **i)** the design of a multiplicative weights update algorithm (in Sec. 3) to construct a generator mixture, and **ii)** a theoretical analysis showing that our generator mixture indeed obtains the pointwise data coverage (1). In fact, we only need a small number of generators to construct the mixture (i.e., it is easy to train), and the distribution $R_{\mathcal{G}}$ for using the mixture is as simple as a uniform distribution (i.e., it is easy to use).

## 2 Related Work

There exists a rich set of works improving classic generative models for alleviating missing modes, especially in the framework of GANs, by altering objective functions [13, 14, 15, 10, 16, 17], changing training methods [18, 19], modifying neural network architectures [2, 20, 21, 22, 23], or regularizing latent space distributions [4, 24]. The general philosophy behind these improvements is to reduce the statistical distance between the generated distribution and target distribution by making

the models easier to train. Despite their technical differences, their optimization goals are all toward reducing a global statistical distance.

The idea of constructing a mixture of generators has been explored, with two ways of construction. In the first way, a set of generators are trained simultaneously. For example, Locatello et al. [25] used multiple generators, each responsible for sampling a subset of data points decided in a k-means clustering fashion. Other methods focus on the use of multiple GANs [26, 27, 28]. The theoretical intuition behind these approaches is by viewing a GAN as a two-player game and extending it to reach a Nash equilibrium with a mixture of generators [26]. In contrast, our method does not depend specifically on GANs, and our game-theoretic view is fundamentally different (recall Sec. 1.1).

Another way of training a mixture of generators takes a sequential approach. This is related to *boosting* algorithms in machine learning. Grnarova et al. [29] viewed the problem of training GANs as finding a mixed strategy in a zero-sum game, and used the Follow-the-Regularized-Leader algorithm [30] for training a mixture of generators iteratively. Inspired by AdaBoost [31], other approaches train a "weak" generator that fits a reweighted data distribution in each iteration, and all iterations together form an additive mixture of generators [32, 33] or a multiplicative mixture of generators [34].

Our method can be also viewed as a boosting strategy. From this perspective, the most related is AdaGAN [33], while significant differences exist. Theoretically, AdaGAN (and other boosting-like algorithms) is based on the assumption that the reweighted data distribution in each iteration becomes progressively easier to learn. It requires a generator in each iteration to have a statistical distance to the reweighted distribution smaller than the previous iteration. As we will discuss in Sec. 5, this assumption is not always feasible. We have no such assumption. Our method can use a weak generator in each iteration. If the generator is more expressive, the theoretical lower bound of our pointwise coverage becomes larger (i.e., a larger $\psi$ in (1)). Algorithmically, our reweighting scheme is simple and different from AdaGAN, only doubling the weights or leaving them unchanged in each iteration. Also, in our mixture of generators, they are treated uniformly, and no mixture weights are needed, whereas AdaGAN needs a set of weights that are heuristically chosen.

To summarize, in stark contrast to all prior methods, our approach is rooted in a different philosophy of training generative models. Rather than striving for reducing a global statistical distance, our method revolves around an explicit notion of complete mode coverage as defined in (1). Unlike other boosting algorithms, our algorithm of constructing the mixture of generators guarantees complete mode coverage, and this guarantee is theoretically proved.

## 3 Algorithm

**A mixture of generators.** Provided a target distribution $P$ on a data domain $\mathcal{X}$, we train a mixture of generators to pursue pointwise mode coverage (1). Let $\boldsymbol{G}^* = \{G_1, \ldots, G_T\}$ denote the resulting mixture of $T$ generators. Each of them ($G_t, t = 1...T$) may use any existing generative model such as GANs. Existing methods that also rely on a mixture of generators associate each generator a nonuniform weight $\alpha_t$ and choose a generator for producing a sample randomly based on the weights. Often, these weights are chosen heuristically, e.g., in AdaGAN [33]. Our mixture is conceptually and computationally simpler. Each generator is treated equally. When using $\boldsymbol{G}^*$ to generate a sample, we first choose a generator $G_i$ uniformly at random, and then use $G_i$ to generate the sample.

**Algorithm overview.** Our algorithm of training $\boldsymbol{G}^*$ can be understood as a specific rule design in the framework of multiplicative weights update [12]. Outlined in Algorithm 1, it runs iteratively. In each iteration, a generator $G_t$ is trained using an updated data distribution $P_t$ (see Line 6-7 of Algorithm 1). The intuition here is simple: if in certain data domain regions the current generator fails to cover the target distribution sufficiently well, then we update the data distribution to emphasize those regions for the next round of generator training (see Line 9 of Algorithm 1). In this way, each generator can focus on the data distribution in individual data regions. Collectively, they are able to cover the distribution over the entire data domain, and thus guarantee pointwise data coverage.

**Training.** Each iteration of our algorithm trains an individual generator $G_t$, for which many existing generative models, such as GANs [9], can be used. The only prerequisite is that $G_t$ needs to be trained to approximate the data distribution $P_t$ moderately well. This requirement arises from our game-theoretic analysis (Sec. 1.1), wherein the total variation distance between $G_t$'s distribution and $P_t$ needs to be upper bounded. Later in our theoretical analysis (Sec. 4), we will formally state this requirement, which, in practice, is easily satisfied by most existing generative models.

---
**Algorithm 1** Constructing a mixture of generators
---
1: **Parameters:** $T$, a positive integer number of generators, and $\delta \in (0, 1)$, a covering threshold.
2: **Input:** a target distribution $P$ on a data domain $\mathcal{X}$.
3: For each $x \in \mathcal{X}$, initialize its weight $w_1(x) = p(x)$.
4: **for** $t = 1 \to T$ **do**
5:     Construct a distribution $P_t$ over $\mathcal{X}$ as follows:
6:     For every $x \in \mathcal{X}$, normalize the probability density $p_t(x) = \frac{w_t(x)}{W_t}$, where $W_t = \int_{\mathcal{X}} w_t(x)\mathrm{d}x$.
7:     Train a generative model $G_t$ on the distribution $P_t$.
8:     Estimate generated density $g_t(x)$ for every $x \in \mathcal{X}$.
9:     For each $x \in \mathcal{X}$, if $g_t(x) < \delta \cdot p(x)$, set $w_{t+1}(x) = 2 \cdot w_t(x)$. Otherwise, set $w_{t+1}(x) = w_t(x)$.
10: **end for**
11: **Output:** a mixture of generators $\boldsymbol{G}^* = \{G_1, \ldots, G_T\}$.
---

**Estimation of generated probability density.** In Line 8 of Algorithm 1, we need to estimate the probability $g_t(x)$ of the current generator sampling a data point $x$. Our estimation follows the idea of adversarial training, similar to AdaGAN [33]. First, we train a discriminator $D_t$ to distinguish between samples from $P_t$ and samples from $G_t$. The optimization objective of $D_t$ is defined as

$$\max_{D_t} \underset{x \sim P_t}{\mathbb{E}} [\log D_t(x)] + \underset{x \sim G_t}{\mathbb{E}} [\log(1 - D_t(x))].$$

Unlike AdaGAN [33], here $P_t$ is the currently updated data distribution, not the original target distribution, and $G_t$ is the generator trained in the current round, not a mixture of generators in all past rounds. As pointed out previously [35, 33], once $D_t$ is optimized, we have $D_t(x) = \frac{p_t(x)}{p_t(x)+g_t(x)}$ for all $x \in \mathcal{X}$, and equivalently $\frac{g_t(x)}{p_t(x)} = \frac{1}{D_t(x)} - 1$. Using this property in Line 9 of Algorithm 1 (for testing the data coverage), we rewrite the condition $g_t(x) < \delta \cdot p(x)$ as

$$\frac{g_t(x)}{p(x)} = \frac{g_t(x)}{p_t(x)} \frac{p_t(x)}{p(x)} = \left( \frac{1}{D_t(x)} - 1 \right) \frac{w_t(x)}{p(x)W_t} < \delta,$$

where the second equality utilize the evaluation of $p_t(x)$ in Line 6 (i.e., $p_t(x) = w_t(x)/W_t$).

Note that if the generators $G_t$ are GANs, then the discriminator of each $G_t$ can be reused as $D_t$ here. Reusing $D_t$ introduces no additional computation. In contrast, AdaGAN [33] always has to train an additional discriminator $D_t$ in each round using the mixture of generators of all past rounds.

**Working with empirical dataset.** In practice, the true data distribution $P$ is often unknown when an empirical dataset $\mathbb{X} = \{x_i\}_{i=1}^n$ is given. Instead, the empirical dataset is considered as $n$ i.i.d. samples drawn from $P$. According to the Glivenko-Cantelli theorem [36], the uniform distribution over $n$ i.i.d. samples from $P$ will converge to $P$ as $n$ approaches to infinity. Therefore, provided the empirical dataset, we do not need to know the probability density $p(x)$ of $P$, as every sample $x_i \in \mathbb{X}$ is considered to have a finite and uniform probability measure. An empirical version of Algorithm 1 and more explanation are presented in the supplementary document (Algorithm 2 and Appendix B).

## 4 Theoretical Analysis

We now provide a theoretical understanding of our algorithm, showing that the pointwise data coverage (1) is indeed obtained. Our analysis also sheds some light on how to choose the parameters of Algorithm 1.

### 4.1 Preliminaries

We first clarify a few notational conventions and introduce two new theoretical notions for our subsequent analysis. Our analysis is in continuous setting; results on discrete datasets follow directly.

**Notation.** Formally, we consider a $d$-dimensional measurable space $(\mathcal{X}, \mathcal{B}(\mathcal{X}))$, where $\mathcal{X}$ is the $d$-dimensional data space, and $\mathcal{B}(\mathcal{X})$ is the Borel $\sigma$-algebra over $\mathcal{X}$ to enable probability measure. We use a capital letter (e.g., $P$) to denote a probability measure on this space. When there is no ambiguity, we also refer them as probability distributions (or distributions). For any subset $\mathcal{S} \in \mathcal{B}(\mathcal{X})$, the probability of $\mathcal{S}$ under $P$ is $P(\mathcal{S}) := \Pr_{x \sim P}[x \in \mathcal{S}]$. We use $G$ to denote a generator. When there is

no ambiguity, $G$ also denotes the distribution of its generated samples. All distributions are assumed absolutely continuous. Their probability density functions (i.e., the derivative with respect to the Lebesgue measure) are referred by their corresponding lowercase letters (e.g., $p(\cdot)$, $q(\cdot)$, and $g(\cdot)$).

Moreover, we use $[n]$ to denote the set $\{1, 2, ..., n\}$, $\mathbb{N}_{>0}$ for the set of all positive integers, and $\mathbb{1}(\mathcal{E})$ for the indicator function whose value is 1 if the event $\mathcal{E}$ happens, and 0 otherwise.

$f$**-divergence.** Widely used in objective functions of training generative models, $f$-divergence is a statistical distance between two distributions. Let $P$ and $Q$ be two distributions over $\mathcal{X}$. Provided a convex function $f$ on $(0, \infty)$ such that $f(1) = 0$, $f$-divergence of $Q$ from $P$ is defined as $D_f(Q \parallel P) \coloneqq \int_{\mathcal{X}} f\left(\frac{q(x)}{p(x)}\right) p(x)\mathrm{d}x$. Various choices of $f$ lead to some commonly used $f$-divergence metrics such as total variation distance $D_{\mathrm{TV}}$, Kullback-Leibler divergence $D_{\mathrm{KL}}$, Hellinger distance $D_{\mathrm{H}}$, and Jensen-Shannon divergence $D_{\mathrm{JS}}$ [35, 37]. Among them, total variation distance is upper bounded by many other $f$-divergences. For instance, $D_{\mathrm{TV}}(Q \parallel P)$ is upper bounded by $\sqrt{\frac{1}{2}D_{\mathrm{KL}}(Q \parallel P)}$, $\sqrt{2}D_{\mathrm{H}}(Q \parallel P)$, and $\sqrt{2D_{\mathrm{JS}}(Q \parallel P)}$, respectively. Thus, if two distributions are close under those $f$-divergence measures, so are they under total variation distance. For this reason, our theoretical analysis is based on the total variation distance.

$\delta$**-cover and** $(\delta, \beta)$**-cover.** We introduce two new notions for analyzing our algorithm. The first is the notion of $\delta$-*cover*. Given a data distribution $P$ over $\mathcal{X}$ and a value $\delta \in (0, 1]$, if a generator $G$ satisfies $g(x) \geq \delta \cdot p(x)$ at a data point $x \in \mathcal{X}$, we say that $x$ is $\delta$-covered by $G$ under distribution $P$. Using this notion, the pointwise mode coverage (1) states that $x$ is $\psi$-covered by $G$ under distribution $P$ for all $x \in \mathcal{X}$. We also extend this notion to a measurable subset $\mathcal{S} \in \mathcal{B}(\mathcal{X})$: we say that $\mathcal{S}$ is $\delta$-covered by $G$ under distribution $P$ if $G(\mathcal{S}) \geq \delta \cdot P(\mathcal{S})$ is satisfied.

Next, consider another distribution $Q$ over $\mathcal{X}$. We say that $G$ can $(\delta, \beta)$-*cover* $(P, Q)$, if the following condition holds:

$$\Pr_{x \sim Q}[x \text{ is } \delta\text{-covered by } G \text{ under distribution} P] \geq \beta. \tag{6}$$

For instance, using this notation, Equation (3) in our game-theoretic analysis states that $G$ can $(0.25, 0.4)$-cover $(P, Q)$.

## 4.2 Guarantee of Pointwise Data Coverage

In each iteration of Algorithm 1, we expect the generator $G_t$ to approximate the given data distribution $P_t$ sufficiently well. We now formalize this expectation and understand its implication. Our intuition is that by finding a property similar to (3), we should be able to establish a pointwise coverage lower bound in a way similar to our analysis in Sec. 1.1. Such a property is given by the following lemma (and proved in Appendix E.1).

**Lemma 1.** *Consider two distributions, $P$ and $Q$, over the data space $\mathcal{X}$, and a generator $G$ producing samples in $\mathcal{X}$. For any $\delta, \gamma \in (0, 1]$, if $D_{TV}(G \parallel Q) \leq \gamma$, then $G$ can $(\delta, 1 - 2\delta - \gamma)$-cover $(P, Q)$.*

Intuitively, when $G$ and $Q$ are identified, $\gamma$ is set. If $\delta$ is reduced, then more data points in $\mathcal{X}$ can be $\delta$-covered by $G$ under $P$. Thus, the probability defined in (6) becomes larger, as reflected by the increasing $1 - 2\delta - \gamma$. On the other hand, consider a fixed $\delta$. As the discrepancy between $G$ and $Q$ becomes larger, $\gamma$ increases. Then, sampling an $x$ according to $Q$ will have a smaller chance to land at a point that is $\delta$-covered by $G$ under $P$, as reflected by the decreasing $1 - 2\delta - \gamma$.

Next, we consider Algorithm 1 and identify a sufficient condition under which the output mixture of generators $\boldsymbol{G}^*$ covers every data point with a lower-bounded guarantee (i.e., our goal (1)). Simply speaking, this sufficient condition is as follows: in each round $t$, the generator $G_t$ is trained such that given an $x$ drawn from distribution $P_t$, the probability of $x$ being $\delta$-covered by $G_t$ under $P$ is also lower bounded. A formal statement is given in the next lemma (proved in Appendix E.2).

**Lemma 2.** *Recall that $T \in \mathbb{N}_{>0}$ and $\delta \in (0, 1)$ are the input parameters of Algorithm 1. For any $\varepsilon \in [0, 1)$ and any measurable subset $\mathcal{S} \in \mathcal{B}(\mathcal{X})$ whose probability measure satisfies $P(\mathcal{S}) \geq 1/2^{\eta T}$ with some $\eta \in (0, 1)$, if in every round $t \in [T]$, $G_t$ can $(\delta, 1 - \varepsilon)$-cover $(P, P_t)$, then the resulting mixture of generators $\boldsymbol{G}^*$ can $(1 - \varepsilon/\ln 2 - \eta)\delta$-cover $\mathcal{S}$ under distribution $P$.*

This lemma is about lower-bounded coverage of a measurable subset $\mathcal{S}$, not a point $x \in \mathcal{X}$. At first sight, it is not of the exact form in (1) (i.e., pointwise $\delta$-coverage). This is because formally speaking it makes no sense to talk about covering probability at a single point (whose measure is zero). But as

$T$ approaches to $\infty$, $\mathcal{S}$ that satisfies $P(\mathcal{S}) \geq 1/2^{\eta T}$ can also approach to a point (and $\eta$ approaches to zero). Thus, Lemma 2 provides a condition for pointwise lower-bounded coverage in the limiting sense. In practice, the provided dataset is always discrete, and the probability measure at each discrete data point is finite. Then, Lemma 2 is indeed a sufficient condition for pointwise lower-bounded coverage.

From Lemma 1, we see that the condition posed by Lemma 2 is indeed satisfied by our algorithm, and combing both lemmas yields our final theorem (proved in Appendix E.3).

**Theorem 1.** *Recall that $T \in \mathbb{N}_{>0}$ and $\delta \in (0,1)$ are the input parameters of Algorithm 1. For any measurable subset $\mathcal{S} \in \mathcal{B}(\mathcal{X})$ whose probability measure satisfies $P(\mathcal{S}) \geq 1/2^{\eta T}$ with some $\eta \in (0,1)$, if in every round $t \in [T]$, $D_{TV}(G_t \parallel P_t) \leq \gamma$, then the resulting mixture of generators $\boldsymbol{G}^*$ can $(1 - (\gamma + 2\delta)/\ln 2 - \eta)\delta$-cover $\mathcal{S}$ under distribution $P$.*

In practice, existing generative models (such as GANs) can approximate $P_t$ sufficiently well, and thus $D_{TV}(G_t \parallel P_t) \leq \gamma$ is always satisfied for some $\gamma$. According to Theorem 1, a pointwise lower-bounded coverage can be obtained by our Algorithm 1. If we choose to use a more expressive generative model (e.g., a GAN with a stronger network architecture), then $G_t$ can better fit $P_t$ in each round, yielding a smaller $\gamma$ used in Theorem 1. Consequently, the pointwise lower bound of the data coverage becomes larger, and effectively the coefficient $\psi$ in (1) becomes larger.

### 4.3 Insights from the Analysis

**$\gamma$, $\eta$, $\delta$, and $T$ in Theorem 1.** In Theorem 1, $\gamma$ depends on the expressive power of the generators being used. It is therefore determined once the generator class $\mathcal{G}$ is chosen. But $\eta$ can be directly set by the user and a smaller $\eta$ demands a larger $T$ to ensure $P(\mathcal{S}) \geq 1/2^{\eta T}$ is satisfied. Once $\gamma$ and $\eta$ is determined, we can choose the best $\delta$ by maximizing the coverage bound (i.e., $(1-(\gamma+2\delta)/\ln 2-\eta)\delta$) in Theorem 1. For example, if $\gamma \leq 0.1, \eta \leq 0.01$, then $\delta \approx 1/4$ would optimize the coverage bound (see Appendix E.4 for more details), and in this case the coefficient $\psi$ in (1) is at least $1/30$.

Theorem 1 also sets the tone for the training cost. As explained in Appendix E.4, given a training dataset of size $n$, the size of the generator mixture, $T$, needs to be at most $O(\log n)$. This theoretical bound is consistent with our experimental results presented in Sec. 5. In practice, only a small number of generators are needed.

**Estimated density function $g_t$.** The analysis in Sec. 4.2 assumes that the generated probability density $g_t$ of the generator $G_t$ in each round is known, while in practice we have to estimate $g_t$ by training a discriminator $D_t$ (recall Section 3). Fortunately, only mild assumptions in terms of the quality of $D_t$ are needed to retain the pointwise lower-bounded coverage. Roughly speaking, $D_t$ needs to meet two conditions: 1) In each round $t$, only a fraction of the covered data points (i.e., those with $g_t(x) \geq \delta \cdot p(x)$) is falsely classified by $D_t$ and doubled their weights. 2) In each round $t$, if the weight of a data point $x$ is not doubled based on the estimation of $D_t(x)$, then there is a good chance that $x$ is truly covered by $G_t$ (i.e., $g_t(x) \geq \delta \cdot p(x)$). A detailed and formal discussion is presented in Appendix E.5. In short, our estimation of $g_t$ would not deteriorate the efficacy of the algorithm, as also confirmed in our experiments.

**Generalization.** An intriguing question for all generative models is their *generalization* performance: how well can a generator trained on an empirical distribution (with a finite number of data samples) generate samples that follow the true data distribution? While the generalization performance has been long studied for supervised classification, generalization of generative models remains a widely open theoretical question. We propose a notion of generalization for our method, and provide a preliminary theoretical analysis. All the details are presented in Appendix E.6.

## 5 Experiments

We now present our major experimental results, while referring to Appendix F for network details and more results. We show that our mixture of generators is able to cover all the modes in various synthetic and real datasets, while existing methods always have some modes missed.

Previous works on generative models used the Inception Score [1] or the Fréchet Inception Distance [18] as their evaluation metric. But we do not use them, because they are both global measures, not reflecting mode coverage in local regions [38]. Moreover, these metrics are designed to measure the quality of generated images, which is orthogonal to our goal. For example, one can always use a more expressive GAN in each iteration of our algorithm to obtain better image quality and thus better inception scores.

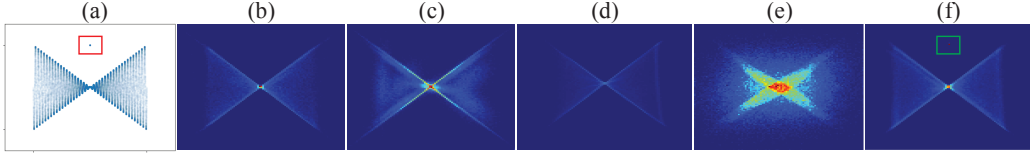

| (a) | (b) | (c) | (d) | (e) | (f) |

Figure 2: **Generative models on synthetic dataset.** (a) The dataset consists of two modes: one major mode as an expanding sine curve ($y = x \sin \frac{4x}{\pi}$) and a minor mode as a Gaussian located at $(10, 0)$ (highlighted in the reb box). (b-f) We show color-coded distributions of generated samples from (b) EM, (c) GAN, (d) AdaGAN, (e) VAE, and (f) our method (i.e., a mixture of GANs). Only our method is able to cover the second mode (highlighted in the green box; zoomin to view).

Since the phenomenon of missing modes is particularly prominent in GANs, our experiments emphasize on the mode coverage performance of GANs and compare our method (using a mixture of GANs) with DCGAN [39], MGAN [27], and AdaGAN. The latter two also use multiple GANs to improve mode coverage, although they do *not* aim for the same mode coverage notion as ours.

**Overview.**   We first outline all our experiments, including those presented in Appendix F. **i)** We compare our method with a number of classic generative models on a synthetic dataset. **ii)** In Appendix F.3, we also compare our method with AdaGAN [33] on other synthetic datasets as well as stacked MNIST dataset, because both are boosting algorithms aiming at improving mode coverage. **iii)** We further compare our method with a single large DCGAN, AdaGAN, and MGAN on the Fashion-MNIST dataset [40] mixed with a very small portion of MNIST dataset [41].

**Various generative models on synthetic dataset.**   As we show in Appendix A, many generative models, such as expectation-maximization (EM) methods, VAEs, and GANs, all rely on a global statistical distance in their training. We therefore test their mode coverage and compare with ours. We construct on $\mathbb{R}^2$ a synthetic dataset with two modes. The first mode consists of data points whose $x$-coordinate is uniformly sampled by $x_i \sim [-10, 10]$ and the $y$-coordinate is $y_i = x_i \sin \frac{4x_i}{\pi}$. The second mode has data points forming a Gaussian at $(0, 10)$. The total number of data points in the first mode is $400\times$ of the second. As shown in Figure 2, generative models include EM, GAN, VAE, and AdaGAN [33] all fail to cover the second mode. Our method, in contrast, captures both modes. We run KDE to estimate the likelihood of our generated samples on our synthetic data experiments (using KDE bandwidth=0.1). We compute $L = 1/N \sum_i P_{model}(x_i)$, where $x_i$ is a sample in the minor mode. For the minor mode, our method has a mean log likelihood of -1.28, while AdaGAN has only -967.64 (almost no samples from AdaGAN).

**Fashion-MNIST and partial MNIST.**   Our next experiment is to challenge different GAN models with a real dataset that has separated and unbalanced modes. This dataset consists of the entire training dataset of Fashion-MNIST (with 60k images) mixed with randomly sampled 100 MNIST images labeled as "1". The size of generator mixture is always set to be 30 for AdaGAN, MGAN and our method, and all generators share the same network structure. Additionally, when comparing with a *single* DCGAN, we ensure that the DCGAN's total number of parameters is comparable to the total number of parameters of the 30 generators in AdaGAN, MGAN, and ours.

|  | "1"s | Frequency | Avg Prob. |
|---|---|---|---|
| DCGAN | 13 | $0.14 \times 10^{-4}$ | 0.49 |
| MGAN | collapsed | - | - |
| AdaGAN | 60 | $0.67 \times 10^{-4}$ | 0.45 |
| Our method | 289 | $3.2 \times 10^{-4}$ | 0.68 |

Table 1: **Ratios of generated images classified as "1".** We generate $9 \times 10^5$ images from each method. The second column indicates the numbers of samples being classified as "1", and the third column indicates the ratio. In the fourth column, we average the prediction probabilities over all generated images that are classified as "1".

To evaluate the results, we train an 11-class classifier to distinguish the 10 classes in Fashion-MNIST and one class in MNIST (i.e., "1"). First, we check how many samples from each method are classified as "1". The test setup and results are shown in Table 1 and its caption. The results suggest that our method can generate more "1" samples with higher prediction confidence. Note that MGAN has a strong mode collapse and fails to produce "1" samples. While DCGAN and AdaGAN generate some samples that are classified as "1", inspecting the generated images reveals that those samples are all visually far from "1"s, but incorrectly classified by the pre-trained classifier (see Figure 3). In contrast, our method is able to generate samples close to "1". We also note that our method can produce higher-quality images if the underlying generative models in each round become stronger.

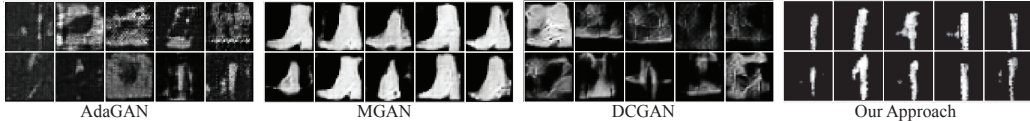

AdaGAN&emsp;&emsp;&emsp;MGAN&emsp;&emsp;&emsp;DCGAN&emsp;&emsp;&emsp;Our Approach

Figure 3: **Most confident "1" samples.** Here we show samples that are generated by each tested methods and also classified by the pre-trained classifier most confidently as "1" images (i.e., top 10 in terms of the classified probability). Samples of our method are visually much closer to "1".

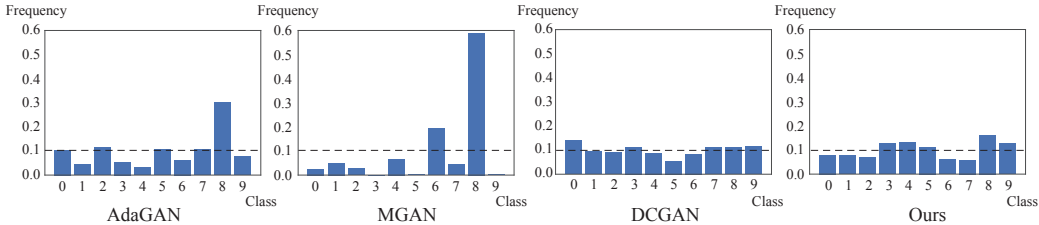

AdaGAN&emsp;&emsp;&emsp;MGAN&emsp;&emsp;&emsp;DCGAN&emsp;&emsp;&emsp;Ours

Figure 5: **Distribution of generated samples.** Training samples are drawn uniformly from each class. But generated samples by AdaGAN and MGAN are considerably nonuniform, while those from DCGAN and our method are more uniform. This experiment suggests that the conventional heuristic of reducing a statistical distance might not merit its use in training generative models.

**Another remarkable feature** is observed in our algorithm. In each round of our training algorithm, we calculate the total weight $\bar{w}_t$ of provided training samples classified as "1" as well as the total weight $W_t$ of all training samples. When plotting the ratio $\bar{w}_t/W_t$ changing with respect to the number of rounds (Figure 4), interestingly, we found that this ratio has a maximum value at around 0.005 in this example. We conjecture that in the training dataset if the ratio of "1" images among all training images is around $1/200$, then a single generator may learn and generate "1" images (the minority mode). To verify this conjecture, we trained a GAN (with the same network structure) on another

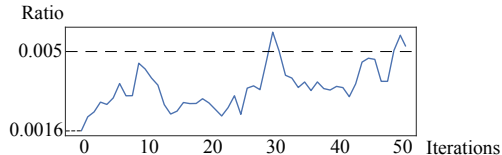

Figure 4: **Weight ratio of "1"s.** We calculate the ratio of the total weights of training images labeled by "1" to the total weights of all training images in each round, and plot here how the ratio changes with respect to the iterations in our algorithm.

training dataset with 60k training images from Fashion-MNIST mixed with 300 MNIST "1" images. We then use the trained generator to sample 100k images. As a result, In a fraction of $4.2 \times 10^{-4}$, those images are classified as "1". Figure 8 in Appendix F shows some of those images. This result confirms our conjecture and suggests that $\bar{w}_t/W_t$ may be used as a measure of *mode bias* in a dataset.

Lastly, in Figure 5, we show the generated distribution over the 10 Fashion-MNIST classes from each tested method. We neglect the class "1", as MGAN fails to generate them. The generated samples of AdaGAN and MGAN is highly nonuniform, though in the training dataset, the 10 classes of images are uniformly distributed. Our method and DCGAN produce more uniform samples. This suggests that although other generative models (such as AdaGAN and MGAN) aim to reduce a global statistical distance, the generated samples may not easily match the empirical distribution—in this case, a uniform distribution. Our method, while not aiming for reducing the statistical distance in the first place, matches the target empirical distribution plausibly, as a byproduct.

# 6 Conclusion

We have presented an algorithm that iteratively trains a mixture of generators, driven by an explicit notion of complete mode coverage. With this notion for designing generative models, our work poses an alternative goal, one that differs from the conventional training philosophy: instead of reducing a global statistical distance between the target distribution and generated distribution, one only needs to make the distance mildly small but not have to reduce it toward a perfect zero, and our method is able to boost the generative model with theoretically guaranteed mode coverage.

**Acknowledgments.** This work was supported in part by the National Science Foundation ( CAREER-1453101, 1816041, 1910839, 1703925, 1421161, 1714818, 1617955, 1740833), Simons Foundation (#491119 to Alexandr Andoni), Google Research Award, a Google PhD Fellowship, a Snap Research Fellowship, a Columbia SEAS CKGSB Fellowship, and SoftBank Group.

## Footnotes

[2]An example of the generator family is the GANs. The definition will be made clear later in this paper.

[3]This requirement is weaker than the mainstream goal of generative models, which all aim to approximate a target data distribution as closely as possible. Here we only require the approximation error is upper bounded.

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
