[Supplementary Material]

# Supplementary Document
# Rethinking Generative Mode Coverage: A Pointwise Guaranteed Approach

## A  Global Statistic Distance Based Generative Approaches

In this section, we analyze a few classic generative models to show their connections to the reduction of a certain global statistical distance. The reliance on global statistical distances explains why they suffer from missing modes, as empirically confirmed in Figure 2 of the main text.

**Maximum Likelihood Estimation.**  Consider a target distribution $P$ with density function $p(\cdot)$. Suppose we are provided with $n$ i.i.d. samples $\{x_1, x_2, \cdots, x_n\}$ drawn from $P$. The goal of training a generator through maximum likelihood estimation (MLE) is to find from a predefined generator family $\mathcal{G}$ the generator $G$ that maximize

$$L(G) = \frac{1}{n} \sum_i \log g(x_i),$$

where $g(\cdot)$ is the probability density function of the distribution generated by $G$. When $n$ approaches $\infty$, the MLE objective amount to

$$\lim_{n \to \infty} \left( \max_{G \in \mathcal{G}} L(G) \right) = \max_{G \in \mathcal{G}} \mathbb{E}_{x \sim P}[\log g(x)] = \max_{G \in \mathcal{G}} \int p(x) \log g(x) \mathrm{d}x = \min_{G \in \mathcal{G}} \left( -\int p(x) \log g(x) \mathrm{d}x \right),$$

which is further equivalent to solve the following optimization problem:

$$\int p(x) \log p(x) \mathrm{d}x + \min_{G \in \mathcal{G}} \left( -\int p(x) \log g(x) \mathrm{d}x \right) = \min_{G \in \mathcal{G}} D_{\mathrm{KL}}(P \parallel G).$$

This is because the first term on the LHS is irrelevant from $G$ and thus is a constant. From this expression, it is evident that the goal of MLE is to minimize a global statistical distance, namely, KL-divergence.

Figure 6 illustrates an 1D example wherein the MLE fails to achieve pointwise coverage. Although Figure 6, for pedagogical purpose, involves a generator family $\mathcal{G}$ consisting of only two generators, it is by no means a pathological case, since in practice generators always have limited expressive power, limited by a number of factors. For GANs, it is limited by the structure of generators. For VAEs, it is the structure of encoders and decoders. For Gaussian Mixture models, it is the dimension of the space and the number of mixture components. Given a $\mathcal{G}$ with limited expressive power, MLE cannot guarantee complete mode coverage.

**Variational Autoencoders (VAEs).**  A VAE has a encoder $\theta \in \Theta$ and a decoder $\phi \in \Phi$ chosen from an encoder and decoder families, $\Theta$ and $\Phi$. It also needs a known prior distribution $Q$ (whose probability density is $q(\cdot)$) of latent variable $z$. Provided a decoder $\phi$ and the prior distribution $Q$, we can construct a generator $G$: to generate an $x$, we firstly sample a latent variable $z \sim Z$ and then sample an $x$ according to the (approximated) likelihood function $p_\phi(x|z)$. To train a VAE, a target distribution $P$ is provided and the training objective is

$$\max_{\theta \in \Theta, \phi \in \Phi} \int_x p(x) \cdot \mathrm{ELBO}_{\theta,\phi}(x) \mathrm{d}x, \tag{7}$$

where $\mathrm{ELBO}_{\theta,\phi}(x)$ is called the evidence lower bound, defined as

$$\mathrm{ELBO}_{\theta,\phi}(x) = \int_z p_\theta(z|x) \log p_\phi(x|z) \mathrm{d}z - \int_z p_\theta(z|x) \log \left( \frac{p_\theta(z|x)}{q(z)} \right) \mathrm{d}z, \tag{8}$$

Here $p_\theta(z|x)$ is the (approximated) posterior function.

Figure 6: Consider a 1D target distribution $P$ with three modes, i.e., a mixture of three Gaussians, $P = 0.98 \cdot \mathcal{N}(0,1) + 0.01 \cdot \mathcal{N}(10,1) + 0.01 \cdot \mathcal{N}(-10,1)$. In this example, the generator class $\mathcal{G}$ only contains two generators. The generated distribution of the first generator $G_1$ is $\mathcal{N}(0,1)$, while the distribution of the second generator $G_2$ is $0.34 \cdot \mathcal{N}(0,1) + 0.33 \cdot \mathcal{N}(10,1) + 0.33 \cdot \mathcal{N}(-10,1)$. In this case, we have $D_{\mathrm{KL}}(P,G_1) \approx 1.28$, $D_{\mathrm{KL}}(P,G_2) \approx 1.40$, $D_{\mathrm{KL}}(G_1,P) \approx 0.029$, and $D_{\mathrm{KL}}(G_2,P) \approx 2.81$ (all $D_{\mathrm{KL}}$ measures use a log base of 2). To minimize $D_{\mathrm{KL}}(P,G)$, maximum likelihood estimation method will choose the first generator, $G_1$. The probability of drawing samples from the side modes (in $[-14, -6]$ and $[6, 14]$) of the target distribution $P$ is $\Pr_{x \sim P}[6 \leq |x| \leq 14] \approx 0.02$, but the probability of generating samples from the first generator in the same intervals is $\Pr_{x \sim G_1}[6 \leq |x| \leq 14] \approx 10^{-9}$. Thus, the side modes are almost missed. To make the first generator satisfy Equation (1), we have to choose $\psi \approx 10^{-7}$, which in practice implies no pointwise coverage guarantee. In contrast, the generated distribution of the second generator can satisfy Equation (1) with $\psi > 1/3$, which is a plausible pointwise coverage guarantee.

Let $G \in \mathcal{G}$ be a generator corresponding to the decoder $\phi$ and the prior $Z$, and let $g(\cdot)$ be the generative probability density of $G$. Then, we have the following derivation:

$$
\begin{aligned}
\mathop{\mathbb{E}}_{x \sim P}[\log g(x)] &= \int_x p(x) \log g(x) \mathrm{d}x = \int_x p(x) \int_z p_\theta(z|x) \log(g(x)) \,\mathrm{d}z \mathrm{d}x \\
&= \int_x p(x) \int_z p_\theta(z|x) \log\left(\frac{p_\phi(x|z)q(z)}{p_\phi(z|x)}\right) \mathrm{d}z \mathrm{d}x \\
&= \int_x p(x) \int_z p_\theta(z|x) \log\left(\frac{p_\phi(x|z)q(z)p_\theta(z|x)}{p_\phi(z|x)p_\theta(z|x)}\right) \mathrm{d}z \mathrm{d}x \\
&= \int_x p(x) \left(\int_z p_\theta(z|x) \log\left(\frac{p_\phi(x|z)q(z)}{p_\theta(z|x)}\right) \mathrm{d}z + \int_z p_\theta(z|x) \log\left(\frac{p_\theta(z|x)}{p_\phi(z|x)}\right) \mathrm{d}z\right) \mathrm{d}x \\
&= \int_x p(x) \left(\int_z p_\theta(z|x) \log\left(\frac{p_\phi(x|z)q(z)}{p_\theta(z|x)}\right) \mathrm{d}z + D_{\mathrm{KL}}\left(p_\theta(z|x) \parallel p_\phi(z|x)\right)\right) \mathrm{d}x \\
&= \int_x p(x) \left(\mathrm{ELBO}_{\theta,\phi}(x) + D_{\mathrm{KL}}\left(p_\theta(z|x) \parallel p_\phi(z|x)\right)\right) \mathrm{d}x.
\end{aligned}
\tag{9}
$$

---
**Algorithm 2** Training on empirical distribution
---
1: **Parameters:** $T$, a positive integer number of generators, and $\delta \in (0, 1)$, a covering threshold.
2: **Input:** a set $\{x_i\}_{i=1}^n$ of i.i.d. samples drawn from an unknown data distribution $P$.
3: For each $x_i$, initialize its weight $w_1(x_i) = 1/n$.
4: **for** $t = 1 \rightarrow T$ **do**
5:     Construct an empirical distribution $\widehat{P}_t$ such that each $x_i$ is drawn with probability $\frac{w_t(x_i)}{W_t}$, where $W_t = \sum_i w_t(x_i)$.
6:     Train $G_t$ on i.i.d. samples drawn from $\widehat{P}_t$.
7:     Train a discriminator $D_t$ to distinguish the samples from $\widehat{P}_t$ and the samples from $G_t$.
8:     For each $x_i$, if $\left(\frac{1}{D_t(x_i)} - 1\right) \cdot \frac{w_t(x_i)}{W_t} < \frac{\delta}{n}$, set $w_{t+1}(x_i) = 2 \cdot w_t(x_i)$.
    Otherwise, set $w_{t+1}(x_i) = w_t(x_i)$.
9: **end for**
10: **Output:** a mixture of generators $\boldsymbol{G}^* = \{G_1, \ldots, G_T\}$.
---

Notice that $D_{\mathrm{KL}}\left(p_\theta(z|x) \parallel p_\phi(z|x)\right)$ is always non-negative and it reaches 0 when $p_\theta(z|x)$ is the same as $p_\phi(z|x)$. This means

$$\mathbb{E}_{x \sim P}[\log g(x)] \geq \int_x p(x) \cdot \mathrm{ELBO}_{\theta,\phi}(x)\mathrm{d}x.$$

If $\theta$ is perfectly trained, i.e., $p_\theta(z|x)$ matches exactly $p_\phi(z|x)$, then

$$\max_{G \in \mathcal{G}} \mathbb{E}_{x \sim P}[\log g(x)] = \max_{\theta \in \Theta, \phi \in \Phi} \int_x p(x) \cdot \mathrm{ELBO}_{\theta,\phi}(x)\mathrm{d}x.$$

From this perspective, it becomes evident that optimizing a VAE essentially amounts to a maximum likelihood estimation. Depending on the generator family $\mathcal{G}$ (determined by $\Phi$ and $Z$) and the encoder family $\Theta$, mode collapse may not always happen. But since it is essentially a maximum likelihood estimation method, the pointwise mode coverage (1) can not be guaranteed in theory, as discussed in the previous paragraph.

**Generative Adversarial Networks (GANs).** Given a target distribution $P$, the objective of training a GAN [9] is to solve the following optimization problem:

$$\min_{G \in \mathcal{G}} \max_D L(G, D),$$

where $L(G, D)$ is defined as

$$L(G, D) = \mathbb{E}_{x \sim P}[\log(D(x))] + \mathbb{E}_{x \sim G}[\log(1 - D(x))] = \int_x p(x)\log(D(x)) + g(x)\log(1 - D(x))\mathrm{d}x.$$

As shown in [9], the optimal discriminator $D^*$ of Nash equilibrium satisfies $D^*(x) \equiv 1/2$. When using $D^*$ in $L(G, D)$, we have

$$L(G, D^*) = D_{\mathrm{KL}}\left(P \parallel \frac{P + G}{2}\right) + D_{\mathrm{KL}}\left(G \parallel \frac{P + G}{2}\right) - 2 = 2D_{\mathrm{JS}}(P \parallel G) - 2,$$

where $D_{\mathrm{JS}}$ is the Jensen-Shannon divergence. Thus, GAN essentially is trying to reduce the global statistical distance, measured by Jensen-Shannon divergence.

There are many variants of GANs, which use (more or less) different loss functions $L(G, D)$ in training. But all of them still focus on reducing a global statistical distance. For example, the loss function of the Wasserstein GAN [10] is $\mathbb{E}_{x \sim P}[D(x)] - \mathbb{E}_{x \sim G}[D(x)]$. Optimizing such a loss function over all 1-Lipschitz $D$ is essentially to reduce the Wasserstein distance, another global statistical distance measure.

## B    Algorithm on Empirical Dataset

In practice, the provided dataset $\{x_i\}_{i=1}^n$ consists of $n$ i.i.d. samples from $P$. According to the Glivenko-Cantelli theorem [36], the uniform distribution over $n$ i.i.d. samples from $P$ will converge

to $P$ when $n$ approaches to infinity. As a simple example, let $P$ be a discrete distribution over two points, $A$ and $B$, with $P(A) = 5/7$ and $P(B) = 2/7$. If 7 samples are drawn from $P$ to form the input data, ideally they should be a multiset $\{A, A, A, A, A, B, B\}$. Each sample has a weight $1/7$, and the total weights of $A$ and $B$ are $5/7$ and $2/7$. Then we will train a generator $G_1$ from the training distribution where point $A$ has training probability $5/7$ and point $B$ has training probability $2/7$.

If the generator $G_1$ obtained is collapsed, e.g., $G_1$ samples $A$ with probability 1 and samples $B$ with probability 0, then ideally the discriminator $D_1$ will satisfy $D_1(A) = 5/12$ and $D_1(B) = 1$. Suppose the parameter $\delta = 1/4$ in Algorithm 1 (and Algorithm 2). We have

$$\left(\frac{1}{D_1(A)} - 1\right) \cdot \frac{w_1(A)}{W_1(A)} = \left(\frac{1}{D_1(A)} - 1\right) \cdot \frac{5}{7} \cdot \frac{1}{5} \geq \delta \cdot P(A) \cdot \frac{1}{5} = \delta/n = \frac{1/4}{7}$$

and

$$\left(\frac{1}{D_1(B)} - 1\right) \cdot \frac{w_1(B)}{W_1(B)} = \left(\frac{1}{D_1(B)} - 1\right) \cdot \frac{2}{7} \cdot \frac{1}{2} < \delta \cdot P(B) \cdot \frac{1}{2} = \delta/n = \frac{1/4}{7}.$$

Thus, each sample $B$ will double the weight, and each sample $A$ will remain the same weight unchanged. The total weight of $A$ is $5/7$, and the total weight of $B$ is $4/7$. In the second iteration, the total probability of $A$ will be decreased to $5/9$ and the total probability of $B$ will be increased to $4/9$. We will use the new probability to train the generator $G_2$ and the discriminator $D_2$, and repeat the above procedure.

In practice, we do not need to know the probability density $p(x)$ of $P$; every sample $x_i$ is considered to have a finite and uniform probability measure. After the generator $G$ is trained over this dataset, its generated sample distribution should approximate well the data distribution $P$. In light of this, the Algorithm 1 can be implemented empirically as what is outlined in Algorithm 2.

## C   Statistical Distance from Lower-bounded Pointwise Coverage

Equation (1) (i.e., $\forall x \in \mathcal{X}, g(x) \geq \psi \cdot p(x)$) is a pointwise lower-bounded data coverage that we pursue in this paper. If Equation (1) is satisfied, then the total variation distance between $P$ and $G$ is automatically upper bounded, because

$$\begin{aligned}
D_{\text{TV}}(P \parallel Q) &= \frac{1}{2} \int_{\mathcal{X}} |p(x) - g(x)| \mathrm{d}x = \int_{\mathcal{X}} \mathbb{1}(p(x) > g(x)) \cdot (p(x) - g(x)) \mathrm{d}x \\
&\leq \int_{\mathcal{X}} \mathbb{1}(p(x) > g(x)) \cdot (p(x) - \psi \cdot p(x)) \mathrm{d}x \\
&= (1 - \psi) \cdot \int_{\mathcal{X}} \mathbb{1}(p(x) > g(x)) \cdot p(x) \mathrm{d}x \\
&\leq 1 - \psi.
\end{aligned}$$

## D   Proof of Equation (3)

Suppose two arbitrary distributions $P$ and $Q$ are defined over a data space $\mathcal{X}$. $G$ is the distribution of generated samples over $\mathcal{X}$. If the total variation distance between $Q$ and $G$ is at most $0.1$, then we have

$$\begin{aligned}
\Pr_{x \sim Q}\left[g(x) \geq \frac{1}{4}p(x)\right] &= \int_{\mathcal{X}} \mathbb{1}\left(g(x) \geq \frac{1}{4}p(x)\right) \cdot q(x) \mathrm{d}x \\
&\geq \int_{\mathcal{X}} \mathbb{1}\left(g(x), q(x) \geq \frac{1}{4}p(x)\right) \cdot q(x) \mathrm{d}x \\
&= \int_{\mathcal{X}} \mathbb{1}\left(q(x) \geq \frac{1}{4}p(x)\right) \cdot q(x) \mathrm{d}x - \int_{\mathcal{X}} \mathbb{1}\left(q(x) \geq \frac{1}{4}p(x) > g(x)\right) \cdot q(x) \mathrm{d}x \\
&\geq \frac{3}{4} - \int_{\mathcal{X}} \mathbb{1}\left(q(x) \geq \frac{1}{4}p(x) > g(x)\right)(q(x) - g(x) + g(x)) \mathrm{d}x \\
&\geq \frac{3}{4} - 0.1 - \frac{1}{4} = 0.4,
\end{aligned}$$

where the first term of the right-hand side of the second inequality follows from

$$\int_{\mathcal{X}} \mathbb{1}\left(q(x) \geq \frac{1}{4}p(x)\right) \cdot q(x)\mathrm{d}x = 1 - \int_{\mathcal{X}} \mathbb{1}\left(q(x) < \frac{1}{4}p(x)\right) \cdot q(x)\mathrm{d}x \geq 1 - \int_{\mathcal{X}} \frac{1}{4}p(x)\mathrm{d}x = \frac{3}{4}.$$

And the third inequality follows from

$$\int_{\mathcal{X}} \mathbb{1}\left(q(x) \geq \frac{1}{4}p(x) > g(x)\right)(q(x) - g(x))\mathrm{d}x \leq \int_{\mathcal{X}} \mathbb{1}(q(x) > g(x))(q(x) - g(x))\mathrm{d}x \leq 0.1,$$

and

$$\int_{\mathcal{X}} \mathbb{1}\left(q(x) > \frac{1}{4}p(x) > g(x)\right)g(x)\mathrm{d}x \leq \int_{\mathcal{X}} \mathbb{1}\left(\frac{1}{4}p(x) > g(x)\right)g(x)\mathrm{d}x \leq \int_{\mathcal{X}} \frac{1}{4}p(x)\mathrm{d}x \leq \frac{1}{4}.$$

# E   Theoretical Analysis Details

In this section, we provide proofs of the lemmas and theorem presented in Section 4. We repeat the statements of the lemmas and theorem before individual proofs. We also provide details to further elaborate the discussion provided in Sec. 4.3 of the paper.

We follow the notations introduced in Sec. 4 of the main text. In addition, we will use $\log(\cdot)$ to denote $\log_2(\cdot)$ for short.

## E.1   Proof of Lemma 1

**Lemma 1.** *Consider two distributions, $P$ and $Q$, over the data space $\mathcal{X}$, and a generator $G$ producing samples in $\mathcal{X}$. For any $\delta, \gamma \in (0, 1]$, if $D_{TV}(G \parallel Q) \leq \gamma$, then $G$ can $(\delta, 1 - 2\delta - \gamma)$-cover $(P, Q)$.*

*Proof.* Since $D_{\mathrm{TV}}(G\|Q) \leq \gamma$ and $\int_{\mathcal{X}} q(x)\mathrm{d}x = \int_{\mathcal{X}} g(x)\mathrm{d}x = 1$, we know that

$$D_{\mathrm{TV}}(G \parallel Q) = \frac{1}{2}\int_{\mathcal{X}} |q(x) - g(x)|\mathrm{d}x = \int_{\mathcal{X}} \mathbb{1}(q(x) > g(x)) \cdot (q(x) - g(x))\mathrm{d}x \leq \gamma. \quad (10)$$

Next, we derive a lower bound of $\Pr_{x \sim Q}[x \text{ is } \delta\text{-covered by } G \text{ under } P]$:

$$\Pr_{x \sim Q}[x \text{ is } \delta\text{-covered by } G \text{ under } P]$$

$$= \int_{\mathcal{X}} \mathbb{1}(g(x) \geq \delta \cdot p(x)) \cdot q(x)\mathrm{d}x \geq \int_{\mathcal{X}} \mathbb{1}(g(x), q(x) \geq \delta \cdot p(x)) \cdot q(x)\mathrm{d}x$$

$$= \int_{\mathcal{X}} \mathbb{1}(q(x) \geq \delta \cdot p(x)) \cdot q(x)\mathrm{d}x - \int_{\mathcal{X}} \mathbb{1}(q(x) \geq \delta \cdot p(x) > g(x)) \cdot q(x)\mathrm{d}x$$

$$= 1 - \int_{\mathcal{X}} \mathbb{1}(q(x) < \delta \cdot p(x)) \cdot q(x)\mathrm{d}x - \int_{\mathcal{X}} \mathbb{1}(q(x) \geq \delta \cdot p(x) > g(x)) \cdot q(x)\mathrm{d}x$$

$$\geq 1 - \delta\int_{\mathcal{X}} p(x)\mathrm{d}x - \int_{\mathcal{X}} \mathbb{1}(q(x) \geq \delta \cdot p(x) > g(x)) \cdot q(x)\mathrm{d}x$$

$$= 1 - \delta - \int_{\mathcal{X}} \mathbb{1}(q(x) \geq \delta \cdot p(x) > g(x)) \cdot (q(x) - g(x) + g(x))\mathrm{d}x$$

$$= 1 - \delta - \int_{\mathcal{X}} \mathbb{1}(q(x) \geq \delta \cdot p(x) > g(x)) \cdot (q(x) - g(x))\mathrm{d}x - \int_{\mathcal{X}} \mathbb{1}(q(x) \geq \delta \cdot p(x) > g(x)) \cdot g(x)\mathrm{d}x$$

$$\geq 1 - \delta - \gamma - \int_{\mathcal{X}} \mathbb{1}(q(x) \geq \delta \cdot p(x) > g(x)) \cdot g(x)\mathrm{d}x$$

$$\geq 1 - \delta - \gamma - \delta\int_{\mathcal{X}} p(x)\mathrm{d}x = 1 - 2\delta - \gamma,$$

where the first equality follows from definition, the second equality follows from $\mathbb{1}(q(x) \geq \delta \cdot p(x)) = \mathbb{1}(g(x), q(x) \geq \delta \cdot p(x)) + \mathbb{1}(q(x) \geq \delta \cdot p(x) > g(x))$, the third inequality follows from Equation (10), and the last inequality follows from

$$\int_{\mathcal{X}} \mathbb{1}(q(x) \geq \delta \cdot p(x) > g(x)) \cdot g(x)\mathrm{d}x \leq \int_{\mathcal{X}} \mathbb{1}(\delta \cdot p(x) > g(x)) \cdot g(x)\mathrm{d}x \leq \int_{\mathcal{X}} \delta \cdot p(x)\mathrm{d}x.$$

$\square$

### E.2 Proof of Lemma 2

Here we first assume that the probability density $g_t$ of generated samples is known. In Appendix E.5, we will consider the case where $g_t$ is estimated by a discriminator as described in Section 3.

**Lemma 2.** *Recall that $T \in \mathbb{N}_{>0}$ and $\delta \in (0,1)$ are the input parameters of Algorithm 1. For any $\varepsilon \in [0,1)$ and any measurable subset $\mathcal{S} \in \mathcal{B}(\mathcal{X})$ whose probability measure satisfies $P(\mathcal{S}) \geq 1/2^{\eta T}$ with some $\eta \in (0,1)$, if in every round $t \in [T]$, $G_t$ can $(\delta, 1-\varepsilon)$-cover $(P, P_t)$, then the resulting mixture of generators $\boldsymbol{G}^*$ can $(1 - \varepsilon/\ln 2 - \eta)\delta$-cover $\mathcal{S}$ under distribution $P$.*

*Proof.* First, we consider the total weight $W_{t+1}$ after $t$ rounds, we derive the following upper bound:

$$
\begin{aligned}
W_{t+1} &= \int_{\mathcal{X}} w_{t+1}(x)\mathrm{d}x = \int_{\mathcal{X}} w_t(x) \cdot (1 + \mathbb{1}(g_t(x) < \delta \cdot p(x)))\mathrm{d}x \\
&= W_t + W_t \cdot \int_{\mathcal{X}} \mathbb{1}(g_t(x) < \delta \cdot p(x)) \cdot \frac{w_t(x)}{W_t}\mathrm{d}x \\
&= W_t + W_t \cdot \int_{\mathcal{X}} \mathbb{1}(g_t(x) < \delta \cdot p(x)) \cdot p_t(x)\mathrm{d}x \\
&= W_t + W_t \cdot \Pr_{x \sim P_t}[g_t(x) < \delta \cdot p(x)] \\
&= W_t + W_t \cdot (1 - \Pr_{x \sim P_t}[g_t(x) \geq \delta \cdot p(x)]) \\
&\leq W_t + W_t \cdot (1 - (1 - \varepsilon)) \\
&\leq W_t \cdot (1 + \varepsilon),
\end{aligned}
$$

where the first equality follows from definition, the second equality follows from Line 9 of Algorithm 1, the forth equality follows from the construction of distribution $P_t$. In addition, the first inequality follows from that $G_t$ can $(\delta, 1-\varepsilon)$-cover $(P, P_t)$. Thus, $W_{T+1} \leq W_1 \cdot (1+\varepsilon)^T = (1+\varepsilon)^T$.

On the other hand, we have

$$
\begin{aligned}
W_{T+1} &= \int_{\mathcal{X}} w_{T+1}(x)\mathrm{d}x \geq \int_{\mathcal{S}} w_{T+1}(x)\mathrm{d}x \geq \int_{\mathcal{S}} 2^{\sum_{t=1}^{T} \mathbb{1}(g_t(x) < \delta \cdot p(x))} p(x)\mathrm{d}x \\
&= \mathbb{E}_{x \sim P}\left[2^{\sum_{t=1}^{T} \mathbb{1}(g_t(x) < \delta \cdot p(x))} \Big| x \in \mathcal{S}\right] \Pr_{x \sim P}[x \in \mathcal{S}],
\end{aligned}
\tag{11}
$$

where the first equality follows from definition, the first inequality follows from $\mathcal{S} \subseteq \mathcal{X}$, and the second inequality follows from Line 9 of Algorithm 1. Dividing both sides by $\Pr_{x \sim P}[x \in \mathcal{S}]$ of (11) and taking the logarithm yield

$$
\begin{aligned}
\log\left(\frac{W_{T+1}}{\Pr_{x \sim P}[x \in \mathcal{S}]}\right) &\geq \log\left(\mathbb{E}_{x \sim P}\left[2^{\sum_{t=1}^{T} \mathbb{1}(g_t(x) < \delta \cdot p(x))} \Big| x \in \mathcal{S}\right]\right) \\
&\geq \mathbb{E}_{x \sim P}\left[\sum_{t=1}^{T} \mathbb{1}(g_t(x) < \delta \cdot p(x)) \Big| x \in \mathcal{S}\right],
\end{aligned}
\tag{12}
$$

where the last inequality follows from Jensen's inequality.

Lastly, we have a lower bound for $\Pr_{x \sim G}[x \in \mathcal{S}]$:

$$
\begin{aligned}
\Pr_{x \sim G}[x \in \mathcal{S}] &= \int_{\mathcal{S}} \frac{1}{T} \sum_{t=1}^{T} g_t(x) \mathrm{d}x \geq \int_{\mathcal{S}} \frac{1}{T} \sum_{t=1}^{T} (\mathbb{1}(g_t(x) \geq \delta \cdot p(x)) \cdot g_t(x)) \mathrm{d}x \\
&\geq \int_{\mathcal{S}} \frac{1}{T} \sum_{t=1}^{T} (\mathbb{1}(g_t(x) \geq \delta \cdot p(x)) \cdot \delta \cdot p(x)) \mathrm{d}x \\
&= \frac{\delta}{T} \int_{\mathcal{S}} \sum_{t=1}^{T} \mathbb{1}(g_t(x) \geq \delta \cdot p(x)) \cdot p(x) \mathrm{d}x \\
&= \frac{\delta}{T} \mathop{\mathbb{E}}_{x \sim P} \left[ \sum_{t=1}^{T} \mathbb{1}(g_t(x) \geq \delta \cdot p(x)) \middle| x \in \mathcal{S} \right] \cdot \Pr_{x \sim P}[x \in \mathcal{S}] \\
&= \frac{\delta}{T} \left( T - \mathop{\mathbb{E}}_{x \sim P} \left[ \sum_{t=1}^{T} \mathbb{1}(g_t(x) < \delta \cdot p(x)) \middle| x \in \mathcal{S} \right] \right) \cdot \Pr_{x \sim P}[x \in \mathcal{S}] \\
&\geq \delta(1 - \log(W_{T+1}/\Pr_{x \sim P}[x \in \mathcal{S}])/T) \cdot \Pr_{x \sim P}[x \in \mathcal{S}] \\
&\geq \delta(1 - \varepsilon/\ln 2 - \eta) \cdot \Pr_{x \sim P}[x \in \mathcal{S}],
\end{aligned}
$$

where the third inequality follows from Equation (12), while the last inequality follows from $\log(W_{T+1}) \leq \log((1+\varepsilon)^T) \leq \varepsilon T/\ln 2$ and $\Pr_{x \sim P}[x \in \mathcal{S}] = P(\mathcal{S}) \geq 1/2^{\eta^T}$. $\qquad\square$

## E.3  Proof of Theorem 1

**Theorem 1.** *Recall that $T \in \mathbb{N}_{>0}$ and $\delta \in (0,1)$ are the input parameters of Algorithm 1. For any measurable subset $\mathcal{S} \in \mathcal{B}(\mathcal{X})$ whose probability measure satisfies $P(\mathcal{S}) \geq 1/2^{\eta^T}$ with some $\eta \in (0,1)$, if in every round $t \in [T]$, $D_{TV}(G_t \parallel P_t) \leq \gamma$, then the resulting mixture of generators $G^*$ can $(1 - (\gamma + 2\delta)/\ln 2 - \eta)\delta$-cover $\mathcal{S}$ under distribution $P$.*

*Proof.* From Lemma 1, we have $\forall t \in [T]$, $G_t$ can $(\delta, 1 - \gamma - 2\delta)$-cover $(P, P_t)$. Combining it with Lemma 2, we have $\forall \mathcal{S} \subseteq \mathcal{X}$ with $P(\mathcal{S}) \geq 1/2^{\eta^T}$, $G$ can $(1 - (\gamma + 2\delta)/\ln 2 - \eta)\delta$-cover $\mathcal{S}$. $\qquad\square$

## E.4  Choice of $T$ and $\delta$ according to Theorem 1

Suppose the empirical dataset has $n$ data points independently sampled from a target distribution $P$. We claim that in our train algorithm, $T = O(\log n)$ suffices. This is because if a subset $\mathcal{S} \in \mathcal{B}(\mathcal{X})$ has a sufficiently small probability measure, for example, $P(\mathcal{S}) < 1/n^3$, then with a high probability (i.e., at least $1 - 1/n^2$), no data samples in $\{x_i\}_{i=1}^{n}$ is located in $\mathcal{S}$. In other words, the empirical dataset of size $n$ reveals almost no information of a subset $\mathcal{S}$ if $P(\mathcal{S}) < 1/n^3$, or equivalently if $1/2^{\eta^T} \approx 1/n^3$ (according to Theorem 1). This shows that $T = O(\log n)$ suffices.

Theorem 1 also sheds some light on the choice of $\delta$ in Algorithm 1 (and Algorithm 2 in practice). We now present the analysis details for choosing $\delta$. We use $\mathcal{G}$ to denote the type of generative models trained in each round of our algorithm. According to Theorem 1, if we know $\eta$ (depends on $T$) and $\gamma$ (depends on $\mathcal{G}$), then we wish to maximize the lower bound $(1 - (\gamma + 2\delta)/\ln 2 - \eta)\delta$ over $\delta$, and the optimal $\delta$ is $\frac{(1-\eta)\ln 2 - \gamma}{4}$. Although in practice $\gamma$ is unknown and not easy to estimate, we note that $\gamma$ is relatively small in practice, and $\eta$ can be also small when we increase the number of rounds $T$.

Given two arbitrary distributions $P$ and $Q$ over $\mathcal{X}$, if the total variation distance between $Q$ and a generated distribution $G$ is at most $\gamma$ (as we discussed in Sec. 1.1 of the main text), then we have

$$
\begin{aligned}
\Pr_{x \sim Q}[g(x) \geq \delta \cdot p(x)] &= \int_{\mathcal{X}} \mathbb{1}(g(x) \geq \delta p(x)) \cdot q(x) \mathrm{d}x \\
&\geq \int_{\mathcal{X}} \mathbb{1}(g(x), q(x) \geq \delta \cdot p(x)) \cdot q(x) \mathrm{d}x \\
&= \int_{\mathcal{X}} \mathbb{1}(q(x) \geq \delta \cdot p(x)) \cdot q(x) \mathrm{d}x - \int_{\mathcal{X}} \mathbb{1}(q(x) \geq \delta \cdot p(x) > g(x)) \cdot q(x) \mathrm{d}x \\
&\geq 1 - \delta - \int_{\mathcal{X}} \mathbb{1}(q(x) \geq \delta \cdot p(x) > g(x))(q(x) - g(x) + g(x)) \mathrm{d}x \\
&\geq 1 - \delta - \gamma - \delta = 1 - 2\delta - \gamma.
\end{aligned}
$$

As discussed in Section 1.1, we can find a mixture of generators satisfying pointwise $(1 - 2\delta - \gamma)\delta$-coverage. Letting $\gamma = 0$, we see that the optimal choice of $\delta$ in this setting is $1/4$. And in this case, $(1 - 2\delta)\delta = 1/8$ is a theoretical bound of the coverage ratio by our algorithm.

### E.5 Use of Estimated Probability Density $g_t$

In Algorithm 1, we use a discriminator $D_t$ to estimate the probability density $g_t$ of generated samples of each generator $G_t$. The discriminator $D_t$ might not be perfectly trained, causing inaccuracy of estimating $g_t$. We show that the pointwise lower-bound in our data coverage is retained if two mild conditions are fulfilled by $D_t$.

1. In each round, only a bounded fraction of covered data points $x$ (i.e., those with $g_t(x) \geq \delta \cdot p(x)$) is falsely classified and their weights are unnecessarily doubled. Concretely, $\forall t \in [T]$, if a sample $x$ is drawn from distribution $P_t$, then the probability of both events—$x$ is $\delta$-covered by $G_t$ under $P$ and $\left(\frac{1}{D_t(x)} - 1\right) \cdot \frac{w_1(x)}{p(x)W_t} < \delta$—happening is bounded by $\varepsilon'$.

2. For any data point $x \in \mathcal{X}$, if in round $t$, the weight of $x$ is not doubled, then with a good chance, $x$ is really $\delta'$-covered, where $\delta'$ can be smaller than $\delta$. Formally, $\forall x \in \mathcal{X}, |\{t \in [T]|g_t(x) \geq \delta' \cdot p(x)\}| \geq \lambda \cdot \left|\left\{t \in [T]\middle|\left(\frac{1}{D_t(x)} - 1\right) \cdot \frac{w_t(x)}{p(x)W_t} \geq \delta\right\}\right|$. Because $\left(\frac{1}{D_t(x)} - 1\right) \cdot \frac{w_t(x)}{p(x)W_t} < \delta$ happens if and only if $w_{t+1}(x) = 2 \cdot w_t(x)$, we use the event $w_{t+1}(x) = 2 \cdot w_t(x)$ as an indicator of the event $\left(\frac{1}{D_t(x)} - 1\right) \cdot \frac{w_t(x)}{p(x)W_t} < \delta$.

If the condition (1) is satisfied, then we are able to upper bound the total weight $W_{T+1}$. Similarly to the proof of Lemma 2, this can be seen from the following derivation:

$$
\begin{aligned}
W_{t+1} &= \int_{\mathcal{X}} w_{t+1}(x) \mathrm{d}x \\
&\leq \int_{\mathcal{X}} w_t(x) \cdot (1 + \mathbb{1}(g_t(x) < \delta \cdot p(x)) + \mathbb{1}(g_t(x) \geq \delta \cdot p(x) \wedge w_{t+1}(x) = 2w_t(x))) \mathrm{d}x \\
&= W_t + W_t \cdot \int_{\mathcal{X}} (\mathbb{1}(g_t(x) < \delta \cdot p(x)) + \mathbb{1}(g_t(x) \geq \delta \cdot p(x) \wedge w_{t+1}(x) = 2w_t(x))) \cdot \frac{w_t(x)}{W_t} \mathrm{d}x \\
&= W_t + W_t \cdot \int_{\mathcal{X}} (\mathbb{1}(g_t(x) < \delta \cdot p(x)) + \mathbb{1}(g_t(x) \geq \delta \cdot p(x) \wedge w_{t+1}(x) = 2w_t(x))) \cdot p_t(x) \mathrm{d}x \\
&= W_t + W_t \cdot \Pr_{x \sim P_t}[g_t(x) < \delta \cdot p(x)] + W_t \cdot \Pr_{x \sim P_t}[g_t(x) \geq \delta \cdot p(x) \wedge w_{t+1}(x) = 2w_t(x)] \\
&\leq W_t + W_t \cdot (1 - \Pr_{x \sim P_t}[g_t(x) \geq \delta \cdot p(x)]) + W_t \cdot \varepsilon' \\
&\leq W_t + W_t \cdot (1 - (1 - \varepsilon)) + W_t \cdot \varepsilon' \\
&\leq W_t \cdot (1 + \varepsilon + \varepsilon'),
\end{aligned}
$$

Thus, the total weight $W_{T+1}$ is bounded by $(1 + \varepsilon + \varepsilon')^T$. Again in parallel to the proof of Lemma 2, we have

$$W_{T+1} = \int_{\mathcal{X}} w_{T+1}(x)\mathrm{d}x \geq \int_{\mathcal{S}} w_{T+1}(x)\mathrm{d}x \geq \int_{\mathcal{S}} 2^{\sum_{t=1}^{T} \mathbb{1}(w_{t+1}(x)=2\cdot w_t(x))} p(x)\mathrm{d}x$$

$$= \mathop{\mathbb{E}}_{x \sim P}\left[2^{\sum_{t=1}^{T} \mathbb{1}(w_{t+1}(x)=2\cdot w_t(x))}\Big| x \in \mathcal{S}\right] \mathop{\mathrm{Pr}}_{x \sim P}[x \in \mathcal{S}].$$

Dividing both sides by $\mathrm{Pr}_{x \sim P}[x \in \mathcal{S}]$ yields

$$\log\left(\frac{W_{T+1}}{\mathrm{Pr}_{x \sim P}[x \in \mathcal{S}]}\right) \geq \log\left(\mathop{\mathbb{E}}_{x \sim P}\left[2^{\sum_{t=1}^{T} \mathbb{1}(w_{t+1}(x)=2w_t(x))}\Big| x \in \mathcal{S}\right]\right)$$

$$\geq \mathop{\mathbb{E}}_{x \sim P}\left[\sum_{t=1}^{T} \mathbb{1}(w_{t+1}(x) = 2w_t(x))\Big| x \in \mathcal{S}\right].$$

Meanwhile, if the condition (2) is satisfied, then

$$\lambda \cdot \left(T - \mathop{\mathbb{E}}_{x \sim P}\left[\sum_{t=1}^{T} \mathbb{1}(w_{t+1}(x) = 2w_t(x))\Big| x \in \mathcal{S}\right]\right) \leq$$

$$T - \mathop{\mathbb{E}}_{x \sim P}\left[\sum_{t=1}^{T} \mathbb{1}(g_t(x) < \delta' \cdot p(x))\Big| x \in \mathcal{S}\right]. \quad (13)$$

Following the proof of Lemma 2, we obtain

$$\mathop{\mathrm{Pr}}_{x \sim G}[x \in \mathcal{S}] = \int_{\mathcal{S}} \frac{1}{T}\sum_{t=1}^{T} g_t(x)\mathrm{d}x \geq \int_{\mathcal{S}} \frac{1}{T}\sum_{t=1}^{T}(\mathbb{1}(g_t(x) \geq \delta' \cdot p(x)) \cdot g_t(x))\mathrm{d}x$$

$$\geq \int_{\mathcal{S}} \frac{1}{T}\sum_{t=1}^{T}(\mathbb{1}(g_t(x) \geq \delta' \cdot p(x)) \cdot \delta' \cdot p(x))\mathrm{d}x$$

$$= \frac{\delta'}{T} \int_{\mathcal{S}} \sum_{t=1}^{T} \mathbb{1}(g_t(x) \geq \delta' \cdot p(x)) \cdot p(x)\mathrm{d}x$$

$$= \frac{\delta'}{T} \mathop{\mathbb{E}}_{x \sim P}\left[\sum_{t=1}^{T} \mathbb{1}(g_t(x) \geq \delta' \cdot p(x))\Big| x \in \mathcal{S}\right] \cdot \mathop{\mathrm{Pr}}_{x \sim P}[x \in \mathcal{S}]$$

$$= \frac{\delta'}{T}\left(T - \mathop{\mathbb{E}}_{x \sim P}\left[\sum_{t=1}^{T} \mathbb{1}(g_t(x) < \delta' \cdot p(x))\Big| x \in \mathcal{S}\right]\right) \cdot \mathop{\mathrm{Pr}}_{x \sim P}[x \in \mathcal{S}]$$

$$\geq \frac{\delta'\lambda}{T}\left(T - \mathop{\mathbb{E}}_{x \sim P}\left[\sum_{t=1}^{T} \mathbb{1}(w_{t+1}(x) = 2w_t(x))\Big| x \in \mathcal{S}\right]\right)$$

$$\geq \delta'\lambda(1 - \log(W_{T+1}/\mathop{\mathrm{Pr}}_{x \sim P}[x \in \mathcal{S}])/T) \cdot \mathop{\mathrm{Pr}}_{x \sim P}[x \in \mathcal{S}]$$

$$\geq \delta'\lambda(1 - (\varepsilon + \varepsilon')/\ln 2 - \eta) \cdot \mathop{\mathrm{Pr}}_{x \sim P}[x \in \mathcal{S}],$$

where the third inequality follows from Equation (13), and other steps are similar to the proof in Lemma 2. By combining with Lemma 1, the final coverage ratio of Theorem 1 with imperfect discriminators $D_t$ should be $(1 - (\gamma + 2\delta + \varepsilon')/\ln 2 - \eta)\delta'\lambda$.

### E.6  Discussion on Generalization

Recently, Arora et al. [26] proposed the *neural net distance* for measuring generalization performance of GANs. However, their metric still relies on a global distance measure of two distributions, not necessarily reflecting the generalization for pointwise coverage.

While a dedicated answer of this theoretical question is beyond the scope of this work, here we propose our notion of generalization and briefly discuss its implication for our algorithm. Provided

a training dataset consisting of $n$ i.i.d. samples $\{x_i\}_{i=1}^n$ drawn from the distribution $P$, we train a mixture of generators $\boldsymbol{G}^*$. Our notion of generalization is defined as $\Pr_{x \sim P}[x \text{ is } \psi\text{-covered by } \boldsymbol{G}^*]$, the probability of $x$ being $\psi$-covered by empirically trained $\boldsymbol{G}^*$ when $x$ is sampled from the true target distribution $P$. A perfect generalization has a value 1 under this notion. We claim that given fixed $T$ rounds of our algorithm and a constant $\varepsilon \in (0,1)$, if $G_t$ in each round is from a family $\mathcal{G}$ of generators (e.g., they are all GANs with the same network architecture), and if $n$ is at least $\Omega(\varepsilon^{-1} T \log |\mathcal{G}|)$, then we have the generalization $\Pr_{x \sim P}[x \text{ is } \psi\text{-covered by } \boldsymbol{G}^*] \geq 1 - \varepsilon$. Here $|\mathcal{G}|$ is the size of *essentially* different generators in $\mathcal{G}$. Next, we elaborate this statement.

**Generalization Analysis.** Our analysis start with a definition of a *family* of generators. In each round of our algorithm, we train a generator $G_t$. We now identify a family of generators from which $G_t$ is trained. In general, a generator $G$ can be viewed as a pair $(f(\cdot), Z)$, where $Z$ is the latent space distribution (or prior distribution) over the latent space $\mathcal{Z}$, and $f(\cdot)$ is a transformation function that maps the latent space $\mathcal{Z}$ to a target data domain $\mathcal{X}$. Let $z$ be a random variable of distribution $Z$. Then, the generated distribution (i.e., distribution of samples generated by $G$) is denoted by the distribution of $f(z)$. For example, for GANs [9] and VAEs [42], $f(\cdot)$ is a function represented by a neural network, and $Z$ is usually a standard Gaussian or mixture of Gaussians.

In light of this, we define a family $\mathcal{G}$ of generators represented by a pair $(\mathcal{F}, Z)$, where $\mathcal{F}$ is a set of functions mapping from $\mathcal{Z}$ to $\mathcal{X}$. For example, in the framework of GANs, $\mathcal{F}$ can be expressed by a neural network with a finite number of parameters which have bounded values. If the input to the neural network (i.e., the latent space) is also bounded, then we are able to apply *net* argument (see e.g., [26]) to find a finite subset $\mathcal{F}' \subset \mathcal{F}$ such that for any $f \in \mathcal{F}$, there exists a function $f' \in \mathcal{F}'$ sufficiently close to $f$. Then the size of $\mathcal{F}'$, denoted by $|\mathcal{F}'|$, can be regarded as the number of "essentially different" functions (or neural networks).

Recall that the generator family $\mathcal{G}$ can be represented by $(\mathcal{F}, Z)$. If the latent space $Z$ is fixed (such as a single Gaussian), then we can define "essentially different" generators in a way similar to the definition of "essentially different" functions in $\mathcal{F}$. If the number of "essentially different" generators from $\mathcal{G}$ is finite, we define the size of $\mathcal{G}$ as $|\mathcal{G}|$.

With this notion, the number of different mixture of generators $\boldsymbol{G}^* = \{G_1, ..., G_T\}$ is at most $|\mathcal{G}|^T$. Consider a uniform mixture $\boldsymbol{G}^*$ of generators, $G_1, G_2, \cdots, G_T \in \mathcal{G}$. If $\Pr_{x \sim P}[x \text{ is } not \text{ } \psi\text{-covered by } \boldsymbol{G}^*] \geq \varepsilon$, then for $n$ i.i.d. samples $x_1, x_2, \cdots, x_n \sim P$, the probability that every $x_i$ is $\psi$-covered by $G$ is at most $(1-\varepsilon)^n$, that is,

$$\Pr_{x_1,...,x_n \sim P} [\text{every } x_1, ..., x_n \text{ is } \psi\text{-covered by } \boldsymbol{G}^*] \leq (1-\varepsilon)^n.$$

Next, by union bound over all possible mixtures $\boldsymbol{G}^*$ that satisfies $\Pr_{x \sim P}[x \text{ is not } \psi\text{-covered by } \boldsymbol{G}^*] \geq \varepsilon$, we have the following probability bound:

$$\Pr_{x_1,...,x_n \sim P} \left[ \exists \boldsymbol{G}^* \text{s.t.} \Pr_{x \sim P} [x \text{ is } not \text{ } \psi\text{-covered by } \boldsymbol{G}^*] \geq \varepsilon \text{ and every } x_1, ..., x_n \text{ is } \psi\text{-covered by } \boldsymbol{G}^* \right]$$
$$\leq (1-\varepsilon)^n |\mathcal{G}|^T. \quad (14)$$

Thus, if $n \geq \Omega(\varepsilon^{-1} T \log |\mathcal{G}|)$, then with a high probability, the inverse of the probability condition above is true, because in this case $(1-\varepsilon)^n$ on the right-hand side of (14) is small—that is, with a high probability, for any mixture $\boldsymbol{G}^*$ that satisfies $\Pr_{x \sim P}[x \text{ is } not \text{ } \psi\text{-covered by } G] \geq \varepsilon$, there must exist a sample $x_i$ such that $x_i$ cannot be $\psi$-covered by $\boldsymbol{G}^*$. The occurrence of this condition implies that if we find a generator mixture $\boldsymbol{G}^*$ that can $\psi$-cover every $x_i$, then $\Pr_{x \sim P}[x \text{ is } \psi\text{-covered by } G] \geq 1-\varepsilon$. In other words, we conclude that if we have $n \geq \Omega(\varepsilon^{-1} T \log |\mathcal{G}|)$ i.i.d. samples $\{x_i\}_{i=1}^n$ drawn from the distribution $P$, and if our algorithm finds a mixture $\boldsymbol{G}^*$ of generators that can $\psi$-cover every $x_i$, then with a high probability, our notion of generalization has $\Pr_{x \sim P}[x \text{ is } \psi\text{-covered by } G] \geq 1-\varepsilon$.

# F  Experiment Details and More Results

## F.1  Network Architecture and Training Hyperparameters.

In our tests, we construct a mixture of GANs. The network architecture of the GANs in show in Table 2 for experiments on synthetic datasets and in Table 3 for real image datasets. All experiments use Adam optimizer [43] with a learning rate of $10^{-3}$, and we set $\beta_1 = 0.5$ and $\beta_2 = 0.999$ with a mini-batch size of 128.

| layer | output size | activation function |
|---|---|---|
| input (dim 10) | 10 | |
| Linear | 32 | ReLU |
| Linear | 32 | ReLU |
| Linear | 2 | |

Table 2: **Network structure** for synthetic data generator.

| layer | output size | kernel size | stride | BN | activation function |
|---|---|---|---|---|---|
| input (dim 100) | $100 \times 1 \times 1$ | | | | |
| Transposed Conv | $512 \times 4 \times 4$ | 4 | 1 | Yes | ReLU |
| Transposed Conv | $256 \times 8 \times 8$ | 4 | 2 | Yes | ReLU |
| Transposed Conv | $128 \times 16 \times 16$ | 4 | 2 | Yes | ReLU |
| Transposed Conv | channel$\times 32 \times 32$ | 4 | 2 | No | Tanh |

Table 3: **Network structure** for image generator. channel=3 for Stacked MNIST and channel=1 for FasionMNIST+MNIST.

## F.2 Additional Experiment Details on Real Data

**Stacked MNIST dataset.** Stacked MNIST is an augmentation of MNIST dataset [13] for evaluating mode collapse. We randomly sample three images from MNIST dataset and stack them in RGB channels of an image. In this way, we construct a dataset of 100k images, each of which has a dimension of $32 \times 32 \times 3$.

**Pre-trained classifier.** For Fashion-MNIST with partial MNIST dataset, we use all the training data of Fashion-MNIST and MNIST to train a 11-class classifier. For stacked MNIST dataset, we train a 10-class classifier on MNIST, and use it as a 1000-class classifier on stacked MNIST (by applying the 10-class MNIST classifier on each color channel). For each experiment, we regard each class as a mode, and use the pre-trained classifier to classify the generated samples into individual modes. After classifying generated samples, we can estimate the generation probability for each mode.

## F.3 Comparison with AdaGAN on Synthetic Dataset and Stacked MNIST

**Mixture of Gaussians and Stacked MNIST.** We conduct experiments on the same synthetic dataset and Stacked MNIST as used in AdaGAN [33]. All synthetic data points are distributed on a 2D plane, consisting of $M$ ($M = 10$) Gaussians uniformly sampled in a squared region $[-15, 15] \times [-15, 15]$, all with the same variance $\sigma_0^2 = 0.05$.

We evaluate our algorithm by checking how many iterations (i.e., the parameter $T$ in Algorithm 1) it takes to cover all modes, and compare it with AdaGAN. A mode is considered covered, if in $N$ generated samples, there exist at least $0.01 \cdot N/M$ samples landed within a distance $3\sigma_0$ away from the mode's center. The experiments on both our algorithms and AdaGAN are repeated 15 times. On this synthetic dataset, both our algorithm and AdaGAN can cover all modes in 2 iterations. For Stacked MNIST, both our method and AdaGAN can cover all modes in 5 iterations.

**More challenging synthetic datasets.** Furthermore, we test our method and AdaGAN on two other synthetic datasets that have more challenging mode distributions. The first one, referred as $D_s$, has 20 modes distributed along a spiral curve (see Figure 7-left). Each mode is a set of points following a Gaussian distribution (with a variance of 1). The center of $i$-th mode ($i = 1..20$) is located at $(\cos(i/3) \cdot i \cdot i, \sin(i/3) \cdot i \cdot i)$. The second dataset, referred as $D_i$, has $21 \times 21 + 1$ modes, among which $21 \cdot 21 = 441$ modes locate on a $[-10, 10] \times [-10, 10]$ uniform grid and one additional mode is isolated at $(100, 100)$ (see Figure 7-right). Each mode is also a set of points under a Gaussian distribution (with a variance of 0.05).

For both datasets, we evaluate how many modes are covered as the number of iterations increases in both our method and AdaGAN. The mode coverage is defined in the same way as in the previous experiment. As shown in Figure 7, our algorithm covers all the modes, and outperforms AdaGAN on both datasets. In terms of efficiency, AdaGAN takes 437 min (25 iterations) and still miss some modes, while our method takes only 134 min (9 iterations) to cover all modes.

Figure 7: **Challenging datasets.** We compare our method with AdaGAN on two datasets (top). (left) Our method covers all modes in $D_s$ dataset with 20 iterations in average. (right) Our method increases the sampled frequency (sampling weights) of the separate mode as the training iteration progresses, whereas AdaGAN increases the sampling frequency of the separated modes. Eventually, AdaGAN can only cover 14 modes in $D_s$ and never cover the separated mode in $D_i$. In contrast, our method successfully covers all modes.

Figure 8: **Sampled "1" images by a single generator.** Based on the observation we draw from Figure 4, we train a single GAN using 60k Fashion-MNIST images together with 300 MNIST "1" images, and the GAN is able to generate images close to "1". Here we show a few generated images from the resulting generator.