[Reviews · NeurIPS 2019]

Reviewer 1



This paper proposes to use a mixture of generators to solve the mode collapse problem. Specifically, for a certain generator, it mainly focuses on the training data which is not covered by previously trained generators. By training a sequence of generators, this paper provides a theoretical analysis that the mixture of generators can cover the whole data distribution. There are three concerns. The first concern lies in the second paragraph. It claims that most divergences are global and are not ideal for promoting mode coverage. It lacks supports, such as related works and experimental results. Besides, several existing works claims that KL divergence can largely solve the mode collapse problem [1][2]. I think these related works should be discussed in the main part. The second part is about speed and scalability. Since the mixture of generators is trained sequentially, what is the time complexity compared to training a simple DCGAN for all these experimental settings, such as in fashion-mnist and toy-data. The third concern is about the experimental setting. For the first toy data in Figure 2, what’s the weight of the minor mode compared to the major mode? In real-world application, will this mode just be treated as noise? On the other hand, I am curious about the likelihood of samples in the minor mode estimated using the kernel density estimator with sufficient enough samples and sufficient small variance. Besides, this paper does not provide any large scale results, such as cifar10 and celebA. [1] Nguyen, et al. "Dual discriminator generative adversarial nets." Advances in Neural Information Processing Systems. 2017. [2] Du, et al. "Learning Implicit Generative Models by Teaching Explicit Ones." arXiv preprint arXiv:1807.03870. # Post-rebuttal comment I have mixed feelings for this paper. On one hand, this paper proposes a new method (i.e., the point-wise convergence) to define the mode collapse problem, which is novel and interesting. And a corresponding algorithm is provided to train a mixture of generators to address the mode collapse problem. On the other hand, the experiments are the main limitation: no experiment on large scale natural images is provided, such as the Cifar10 and celebA. Indeed, the authors address most of my concerns. However, the claim of the Figure 6 in Appendix B is incorrect. If we use the KL divergence, i.e., KL(P||G), rather than the inverse KL, i.e., KL(G||P), the green distribution will be chosen rather than the red one. According to my numerical results, KL(P||N(0, 1)) = 4.6, whereas KL(P||0.25N(-10, 1) + 0.5N(0, 1) + 0.25N(10, 1)) = 0.36. It verifies the zero-avoiding properties of KL divergence[2]. To summarize, I still have an inclination to reject this paper in its current state, because of the limitation of experiments and incorrect claims for the KL divergence. However, based on its novel contribution, I will also be happy if this paper will be accepted.

Reviewer 2



This paper considers the problem of learning a generator mixture to capture all the data modes under an interesting formulation. Different from existing work which is mainly based on global statistical distance, this work focuses on the local pointwise coverage which also guarantees of the global distance in a loose way. Based on the multiplicative weights update algorithm, the authors proposed an algorithm to learn the generator mixture and theoretically proved its complete mode coverage guarantee. The proposed method works well on both synthetic and real datasets, achieving better mode coverage compared to existing methods while maintaining good learning of the global distribution. On the other hand, the manuscript is not very easy to follow, especially Section 1.1 and Section 3. The introduction of the (\delta,\beta)-cover is not very intuitive even though it would eventually become clear in Section 3.2. Some intuitive discussion of the motivations might be helpful. The proposed method, Algorithm 1, has a flavor of the non-adaptive boosting algorithm where the weight of data points would be reduced if we got thing right and an simple average of the individual weak leaners is outputted. The existing boosted density estimation methods, additive (AdaGAN) or multiplicative ([1,2]), all have a flavor of the adaptive boosting algorithm. It is interesting to see if the adaptive version has certain guarantee in terms of the introduced mode coverage or examples of the failure cases. The empirical results suggest that the proposed method performs better than AdaGAN in terms of both mode coverage (in Table 1) and global distribution matching (in Figure 5). But AdaGAN is directly trying to minimize the global statistical distance while the proposed method only has a loose guarantee in the global sense, it would be better if the authors could provide some insights here. The experiments are relatively weak. The authors proposed a mode coverage metric and checked the global distribution matching all based on trained classifiers which is somewhat ad-hoc, and it would be good to also look at the coverage metric introduced in the AdaGAN work. The precision and recall metrics introduced in [3] are another interesting candidates. There is only mode coverage results for a fixed T value. Plots of the coverage metric against T might shed more light about the learning process. Another question is about the mode bias measure introduced towards the end of Section 4. It could be computed in Figure 4 mainly because the mode itself is known, but it is unclear how to compute this measure in practice where the modes are generally unknown beforehand. [1] Grover, A. and Ermon, S. Boosted generative models. In AAAI, 2018. [2] Cranko, Z., and Nock, R. Boosted density estimation remastered. In ICML, 2019. [3] Sajjadi, M.S.M., Bachem, O., Lucic, M., Bousquet, O., and Gelly, S. Assessing generative models via precision and recall. In NeurIPS, 2018. After rebuttal: Thanks for the detailed rebuttal. I do not fully agree with the argument provided for why AdaGAN can not guarantee full mode coverage. "In the next iteration, the weight of the major 25 mode may still be large, and thus prevents AdaGAN from learning the minor mode". AdaGAN would also diminish the weight for the well-fitted part in an exponential way similar to the proposed method. As pointed out by other reviewers, the quantitative results are limited, especially on real data. But I would maintain my score as 7 since I do think the theoretical contribution is novel and interesting.

Reviewer 3



The paper describes a new algorithm for training a mixture of generator that can, with some theoretical guarantee, capture multiple modes in the data distribution. One of the main contributions of the paper is a new measure of distance between distributions that explicitly considers point-wise coverage (rather than the traditional global notions of coverage, e.g. f-divergence), optimizing for this metric allows one to construct mixtures that aim to capture multiple modes (even minor ones) in the target distribution. It appears that existing work training generators with modified objective functions and those that ensemble multiple generators do not explicitly train for this type of "point-wise" or "local" coverage, but rather train for some proxy that may indirectly encourage coverage of multiple modes. In this light, the contribution of both the distance measure and the algorithm is significant. Furthermore, the proposed algorithm comes with theoretical guarantees of certain degrees of point-wise coverage (given appropriate choices of the number of expressive generators in the mixtures and a covering threshold). This is valuable as the author demonstrate that theoretical analysis lends insight to how to choose hyper parameters in practice. Finally, the paper presents evidence, in cases where the number of ground truth modes are known, for the effectiveness of the proposed method on synthetic and image datasets, and comparison results with respect to other methods that aim to capture multiple modes. Overall the paper is very well written. The technical exposition is clear and easy to follow. The theoretical analysis is thorough and the central idea is intuitive and appealing.

[Author Response · NeurIPS 2019]

**Response to Reviewer #1**

[Support for the claims that most divergences are global and not ideal for mode coverage]: We do provide support for our claims, both theoretically and empirically. In Appendix B, we analyze several classic generative models and show that they all reduce certain statistical distances (such as KL divergence and JS divergence) measured *globally* over the data space. In Fig. 1, we give an intuitive example illustrating why global statistical distances cannot guarantee mode coverage. This claim is further supported by our experiments in Fig. 2. We also note that while some existing generative models can avoid missing modes well in certain situations, it is unclear whether they guarantee a complete mode coverage—a condition that we explicitly define in Eq. (1) and guarantee in our approach.

[Speed and scalability]: As discussed in Section 3.3 and Appendix F.4, the number of generators needed is at most $O(\log n)$. This theoretical bound implies that the running time will be scaled by a factor of at most $O(\log n)$. In practice, this factor is small. In our experiments on synthetic data, AdaGAN takes 437 min (25 iterations) and still miss some modes, while our method takes only 134 min (9 iterations) to cover all modes. In the Stacked MNIST experiment, our method takes 320 min while AdaGAN takes 550 min. Detailed setup can be found in Appendix G.

[Weight of the minor mode]: As presented in Line 305, in Fig. 2, the ratio of the minor mode weight to the major mode weight is $1/400$. In many real-world applications, such an imbalanced mode may be caused by the difficulty of data collection, not necessarily noise. In fact, the currently active research in machine learning fairness is largely motivated by this kind of data imbalance, indicating that the imbalanced modes do exist in real-world datasets.

[KDE]: We run KDE to estimate the likelihood of our generated samples on our synthetic data experiments (using KDE bandwidth=0.1). We compute $L = 1/N \sum_i P_{model}(x_i)$, where $x_i$ is a sample in the minor mode. For the minor mode, our method has a mean log likelihood of -1.28, while AdaGAN has only -967.64 (almost no samples from AdaGAN).

**Response to Reviewer #2.**

[Insight of why AdaGAN can not guarantee full mode coverage]: AdaGAN aims to fit the "residual" distribution in each iteration. As a thought experiment, consider a dataset that has a major mode and a minor mode. In one iteration, AdaGAN may cover the major mode but not perfectly fit its distribution. In the next iteration, the weight of the major mode may still be large, and thus prevents AdaGAN from learning the minor mode. In contrast, if the major mode is covered by our algorithm in an iteration, in the next iteration the relative weight of the major mode (w.r.t. the minor mode weight) will become lower. Eventually, our algorithm learns the minor mode distribution.

[Metrics used in AdaGAN]: We tested our algorithm on synthetic data using the metrics introduced by AdaGAN. We would like to emphasize that the metrics used by AdaGAN are all global measures, different from our focus on *local* mode coverage. The first metric in AdaGAN is $C := P_d(dP_{model} > t)$ with a $t$ satisfying $P_{model}(dP_{model} > t) = 0.95$. We use KDE to estimate $P_{model}$. Over the entire dataset, the score for AdaGAN is 0.931, slightly better than ours 0.872. However, for the scores over the minor mode, we have $C = 1.0$ while AdaGAN has $C = 0.0$ (no samples on it). Another AdaGAN metric is the likelihood of the true data under the generated distribution, $L := 1/N \sum_i P_{model}(x_i)$. AdaGAN has $L = -7.43$, lower than ours $L = -5.03$, because the minor mode is missed in AdaGAN.

**Response to Reviewer #3.**

[Related work]: We will add a section to discuss the major related work in the main text of the revised paper.

[Compare with AdaGAN on synthetic dataset]: We emphasize our comparison with AdaGAN because it has the most similar setting to ours; both are boosting algorithms. Also, in the AdaGAN paper, the authors have compared AdaGAN to several baseline methods, and shown that AdaGAN outperforms other methods in both synthetic and real datasets. Thus, our rationale is to use AdaGAN as our baseline for comparison. Furthermore, it is unclear for us how to change statistical distance or hack the objective function to improve mode coverage guarantee. Such a change itself might merit further research. When training MGAN over our synthetic dataset, we found that it cannot converge and produces poor mode coverage, even after experimenting with multiple parameter configurations. Thus, we choose not to include MGAN in the synthetic data comparisons.

[Comparison with methods using different priors over the latent space]: It is indeed interesting to compare with methods that use different latent space priors. However, those methods lack the theoretical guarantee of complete mode coverage, while our method can provably cover all the modes under our metric. The use of latent space prior can help improving mode coverage, but to our knowledge, how to construct the latent space prior in general is still an open problem and may require task-specific solutions. We will discuss the differences from those methods when we revise our paper.

[Pointwise coverage without ground truth knowledge]: Indeed, it is often hard to know the mode distribution in a dataset. This is precisely the motivation of introducing Eq. (1) as our definition of complete mode coverage (see Line36-39). If the pointwise coverage Eq. (1) is satisfied, modes are guaranteed to be covered, no mater how the modes are defined and distributed. So we don't need to know the mode distribution in order to guarantee mode coverage. When evaluating our method, we need a classifier (under a certain mode definition) to determine whether a mode is covered and quantify mode coverage. For example, in stacked-MNIST, we use the standard MNIST classifier for each color channel and we combine these three classifiers as a 1000-class classifier to quantify the coverage of 1000 modes.

[Meta-Review · NeurIPS 2019]

This paper received good scores overall, with a bit of disparity between reviews. There is consensus that it makes novel contributions in the field latent-variable generative models, providing a metric for mode-collapse and offering a potential solution via a mixture of generators. It also provides a theoretical analysis. The weaknesses of this submission are the experiments and related works, which could be improved to substantially increase the strength of the paper.